# Kriging-based surrogate data-enriching artificial neural network prediction of strength and permeability of permeable cement-stabilized base

Xiaoming Wang[1], Yuanjie Xiao [1,2] ✉, Wenqi Li[1], Meng Wang[1], Yanbin Zhou[3], Yuliang Chen[4] & Zhiyong Li[4]

Limited test data hinder the accurate prediction of mechanical strength and permeability of permeable cement-stabilized base materials (PCBM). Here we show a kriging-based surrogate model assisted artificial neural network (KS-ANN) framework that integrates laboratory testing, mathematical modeling, and machine learning. A statistical distribution model was established from limited test data to enrich the dataset through the combination of markov chain monte carlo simulation and kriging-based surrogate modeling. Subsequently, an artificial neural network (ANN) model was trained using the enriched dataset. The results demonstrate that the well-trained KS-ANN model effectively captures the actual data distribution characteristics. The accurate prediction of the mechanical strength and permeability of PCBM under the constraint of limited data validates the effectiveness of the proposed framework. As compared to traditional ANN models, the KS-ANN model improves the prediction accuracy of PCBM's mechanical strength by 21%. Based on the accurate prediction of PCBM's mechanical strength and permeability by the KS-ANN model, an optimization function was developed to determine the optimal cement content and compaction force range of PCBM, enabling it to concurrently satisfy the requirements of mechanical strength and permeability. This study provides a cost-effective and rapid solution for evaluating the performance and optimizing the design of PCBM and similar materials.

The accelerated progress of artificial intelligence has facilitated the widespread implementation of data-driven machine learning (ML) techniques in material design optimization[1]. Charrier & Ouellet-Plamondon[2] utilized artificial neural networks (ANNs) to evaluate the impact of admixtures on the fresh properties of cement slurries. To optimize the mechanical properties of cement paste, they determined the dosage of each admixture based on the critical yield stress. Other studies have also utilized ANNs[3–5]. Moreover, ANN models have been utilized to predict the physical properties of concrete materials, including durability[6], rheology[7], optimal grading[8], and flexural strength[9]. ANN models have exhibited strong robustness in previous applied studies and effectively solve highly nonlinear problems. However, ANN models require large amounts of training data[10], making them cost-prohibitive in the civil engineering

[1]School of Civil Engineering, Central South University, Changsha, China. [2]Ministry of Education (MOE) Key Laboratory of Engineering Structures of Heavy Haul Railway (Central South University), Changsha, China. [3]The Second Xiangya Hospital of Central South University, Changsha, China. [4]Hunan Communications Research Institute CO., LTD., Changsha, China. ✉e-mail: yjxiao@csu.edu.cn

field, which is primarily based on laboratory experiments. The proxy model is a simplified function approximation of a complex model and can significantly reduce experimental costs, thereby promoting engineering analysis of complex systems[11]. The kriging-based surrogate (KS) model has exhibited powerful analytical ability in predicting the failure probability[12–15], identifying damages in civil engineering structures[16], and significantly reducing experimental costs[16–18]. Therefore, a kriging-based surrogate model needs to be developed to aid ANN models in improving the prediction accuracy, thus reducing experimental costs. Further, the popularity of transfer learning has led to the need for a large number of raw datasets to train valuable weight files of transfer learning[19–22]. However, obtaining such datasets is often challenging in the civil engineering field due to the time-consuming and labor-intensive nature of extensive complex laboratory tests involved. In contrast, a machine learning paradigm based on kriging-based surrogate modeling-aided artificial neural networks can be used to enrich limited datasets. This approach provides valuable references for addressing this challenge and effectively reduces the cost of constructing weighting models of migration learning.

Permeable cement-stabilized base materials (PCBMs) typically have highly interconnected porosity, which allows water to flow smoothly through them under the influence of gravity[23,24]. Although these properties are advantageous in terms of fully permeable road base structures, they have a disadvantage, i.e., decreased mechanical strength. In fact, as the porosity increases, the contact area between cement bridges and the adjacent aggregates decreases, and the cement bridges between aggregates exhibit higher effective stress, which leads to failure at lower load levels, resulting in damage such as subgrade failure, clogged permeable pores, and uneven settlement of the subgrade[25–27]. Therefore, to meet the long-term stable operation requirements of permeable bases, PCBMs with balanced mechanical strength and permeability coefficient are needed. The cement content of PCBMs and the compaction force during construction affect this balance due to changes in the spatial distribution and pore size[28,29]. Due to the variations in the application environments and available materials, the strength and coefficient of permeability of PCBMs under different combinations of cement content and compaction force need to be evaluated based on past experiences and extensive trial-and-error tests. This often entails very high costs as well as repetitive and tedious work, which is detrimental to the promotion and improvement of PCBMs.

PCBM is similar to pervious concrete in terms of the fact that it also confronts challenges in balancing mechanical strength and permeability properties during the design phase. Existing studies have focused mainly on revealing the correlation mechanisms between the compressive strength of pervious concrete and its pervious pore structures and on developing predictive models that relate strength properties to pervious pore characteristics. Such work has proven valuable in guiding the optimal design of pervious concrete[24,30–36]. However, laboratory testing methods are known to be time-consuming, labor-intensive, and financially costly. Furthermore, when changes occur in actual engineering conditions, extensive laboratory tests are required to re-evaluate the hydro-mechanical properties of materials, causing significant financial and labor investments. In light of this deficiency, a variety of alternative methods have been employed to evaluate the mechanical strength and infiltration properties of pervious concrete, including the random lattice discrete particle method[37], discrete element method (DEM)[38–40], finite element method (FEM)[41,42], computational fluid dynamics (CFD) method[43], and other self-developed numerical simulation methods[44–46]. The aim of these methods is to reduce the expenses associated with laboratory tests and expedite the material design process. The balance between the mechanical strength and permeability properties of pervious concrete is a crucial area of

study. By utilizing data-driven machine learning-based methods, it becomes possible to efficiently evaluate these properties and minimize testing costs while enhancing design efficiency. Currently, machine learning models are primarily trained by extensive laboratory test data to expedite the development of high-performance pervious concrete designs[47,48]. However, the development of machine learning models requires substantial datasets, leading to much greater data demands and economic costs. To overcome these challenges, there is a need to create a cost-effective machine-learning model that can streamline experimentation and expedite the design of materials.

As compared to pervious concrete, PCBMs involve more intricate preparation processes and internal structures. Pervious concrete generally contains 20%–40% cement[49,50] and is self-compacted or compacted by lower compaction forces[46,47,51]. Due to its relatively higher cement content, pervious concrete usually exhibits greater mechanical strength. Conversely, PCBM typically contains 3% to 15% cement[52] and is compacted by forces up to 300 kN. A reduced cement content leads to decreased mechanical strength, while increased compaction force may result in inadequate permeability of the PCBM. Xiao et al.[53] established a preliminary linkage between field and laboratory material compaction by directly applying field compaction force levels in laboratory specimen compaction[53]. However, there is still a lack of in-depth research in this area, especially for permeable cement-stabilized base materials. Compared to pervious concrete, PCBMs exhibit a much more intricate balance between mechanical strength and permeability properties. The prediction of PCBM properties, such as mechanical strength, porosity, and coefficient of permeability, from laboratory design parameters (e.g., cement content and compaction force) is highly indispensable for optimizing PCBM design. However, few studies have focused on this specific aspect.

In this study, a kriging-based surrogate data-enriching artificial neural network (KS-ANN) model is developed to achieve high-precision prediction results with a small amount of data and provide heuristic theoretical references for the design optimization and compaction scheme determination of PCBMs. To achieve this main objective, a series of subsidiary studies are conducted. This includes (i) establishing a reliable laboratory test plan to obtain authentic test samples and provide reliable data input for the kriging-based surrogate (KS) models, (ii) using KS models to aid ANN models in effectively predicting the uniaxial compressive strength and coefficient of permeability of PCBMs, and (iii) applying the proposed method to optimize the PCBM design and compaction schemes. The laboratory test of PCBMs is chosen as a prime example. The suggested methodology is integrated into the decision-making process to enhance the design and on-site compaction plan of PCBMs. The suggested approach has the potential to significantly decrease the quantity of laboratory testing specimens as well as testing expenses. Additionally, a connection is established among laboratory testing, mathematical models, and ML, providing useful theoretical insights for projecting the functionality, refining the design, and engineering applications of analogous materials. It's worth noting that the novelty of the study lies in the application of data enrichment techniques combined with machine learning models for the specific PCBMs, rather than purely developing a new technique that can be applied to various types of data problems.

## Results
### Experimental design and results
For PCBMs, the materials are required to have a 28-day unconfined compressive strength of at least 3.5 MPa and a coeffcient of permeability of at least $0.5 \, mm \cdot s^{-1}$[54]. Laboratory uniaxial compression tests are typically performed to directly examine whether the unconfined compressive strength ($\sigma$) of a material meets the usage requirements. However, in the on-site compaction construction procedure, the PCBM is first thoroughly mixed and evenly spread in the subgrade.

Then, static pre-compaction is performed using a 10-ton (approximately 100 kN) steel wheel roller. Subsequently, a second static compaction of the PCBM is performed using a 20-ton (approximately 200 kN) steel wheel roller, resulting in a cement-stabilized base in a well-compacted state. Finally, to achieve the required density, a third static compaction is performed on the cement-stabilized base using a 30-ton (approximately 300 kN) rubber wheel roller. During the three rounds of compaction operations, to ensure that the cement paste adhering to the aggregate surfaces does not fall off, the roller compactor always maintains static compaction mode, which eliminates the vibration compaction process. In laboratory experiments, to reproduce the compacted state of PCBMs, the static pressure method is usually used for sample preparation. To investigate the influences of compaction force and cement content on the strength and coefficient of permeability of PCBMs, this study designed three different levels of static compaction force (i.e., 100 kN, 150 kN, and 200 kN) and three different levels of cement content (i.e., 5%, 10%, and 20%). Therefore, nine different working conditions corresponding to different combinations of cement content and compaction force were specified. The combinations of these parameters yielded the tested unconfined compressive strength (σ), porosity (P), and coefficient of permeability (K), which formed a relatively small dataset. The obtained small datasets were used as inputs for the KS-ANN model, which was trained to predict the strength and coefficient of permeability of PCBMs under the compaction force of 300 kN. Among them, the orthogonal experimental design is shown in Tables 1 where 3 replicate specimens were tested for each group of experiments. The detailed laboratory testing process and analysis of the PCBM test results are shown in the supplementary information document.

In actual engineering applications, to achieve desired permeability, aggregates larger than 4.75 mm are typically used. Figure 1a displays the typical gradation commonly used in actual engineering applications.

The inner diameter and height of the mold used for the unconfined compressive strength test in the laboratory are 140 and 180 mm, respectively. The designed aggregate quantity is 4.0 kg, and the main

rock of the aggregates is limestone with a natural density of approximately 2600 kg/m³. In order to maximize the adhesion of the cement paste to the aggregate surfaces, the aggregates were washed before sample preparation. This is done to prevent impurities, such as dust and soil, from affecting the adhesion rate of the cement paste to the aggregate surfaces. The washed aggregates were then kept in a sealed plastic bag for 24 h before sample preparation. In addition, the adhesion of the cement paste to the aggregate surfaces can be improved by reducing the fluidity of the paste. Thus, the range of aggregate sizes used in this paper is fixed from 4.75–26.5 mm, and the water-cement ratio is fixed at 0.4. The detailed design parameters of the cement paste are shown in Table 2.

After thoroughly mixing and blending the aggregates, cement, and water, they are placed into a mold and compacted using the designed compaction force. After compaction, the sample was de-molded after being left to stand for 24 h under laboratory conditions and then placed in a standard curing room for 28 days. In this study, the porosity (P), coefficient of permeability (K), and unconfined compressive strength (σ) of PCBM specimens were determined to obtain accurate laboratory data. Among them, each combination of conditions included 3 replicate specimens. The unconfined compressive strength (σ), porosity (P), and coefficient of permeability (K) under different combination conditions include a total of 81 data points. The laboratory test results are shown in Table 3.

In practical engineering design, PCBMs need to fulfill the requirements of both unconfined compressive strength and permeability. It is not economical to ensure the balance range of the mechanical strength and permeability of PCBMs through a large number of laboratory experiments, and the application of machine learning methods in databases with few samples mainly suffers from low accuracy and underfitting problems. Therefore, we developed a KS-ANN method to seek to improve the prediction accuracy with a small number of samples and to optimize material design and sample preparation methods (or on-site compaction).

## Correlation analysis of the unconfined compressive strength, porosity, permeability coefficient, compaction force, and cement content of PCBMs

Figure 1b presents the results of the correlation analysis between different design variables and performance indicators. As shown in the figure, the coefficient of permeability (K) and porosity (P) exhibit negative correlations with the compaction force or cement content. The cement content significantly affects the coefficient of permeability (K), with a correlation coefficient of −0.89. The correlation coefficient between the compaction force and coefficient of permeability (K) is only −0.42. The correlation coefficient between the porosity (P) and

### Table 1 | Orthogonal design parameters for the laboratory unconfined compression tests

| Static compaction force (kN) | Cement content (%) | | |
|---|---|---|---|
| | 5 | 10 | 20 |
| 100 | 3 | 3 | 3 |
| 150 | 3 | 3 | 3 |
| 200 | 3 | 3 | 3 |

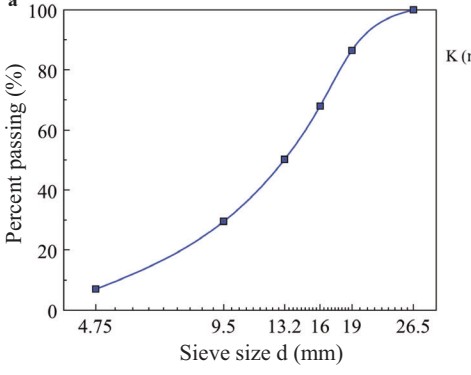

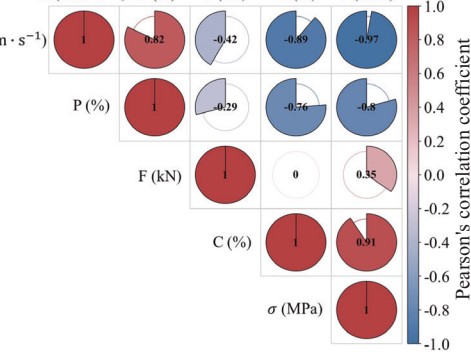

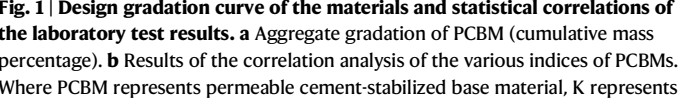

**Fig. 1 | Design gradation curve of the materials and statistical correlations of the laboratory test results. a** Aggregate gradation of PCBM (cumulative mass percentage). **b** Results of the correlation analysis of the various indices of PCBMs. Where PCBM represents permeable cement-stabilized base material, K represents

the coefficient of permeability, P represents the porosity, F represents the compaction force, C represents the cement content, and σ represents the unconfined compressive strength.

cement content is −0.76, while that between the porosity (P) and compaction force is only −0.29. The unconfined compressive strength (σ) of the sample is positively correlated with both the compaction force and cement content, and the unconfined compressive strength (σ) has a significantly greater correlation coefficient with the cement content (0.91) than with the compaction force (0.35). Additionally, the correlation coefficient between the compaction force and cement content is 0, indicating no correlation. However, the values of the unconfined compressive strength (σ), porosity (P), and coefficient of permeability (K) differ under different combinations of compaction force and cement content. Therefore, significant statistical inter-correlations exist among the unconfined compressive strength (σ), porosity (P), coefficient of permeability (K), and sample preparation conditions, which is a prerequisite for using a kriging-based surrogate model.

The above analysis indicates that an excessively high compaction force (corresponding to the on-site compaction force) has less effect on the unconfined compressive strength of PCBMs, whereas a slight increase in cement content can significantly enhance the unconfined compressive strength of PCBMs. However, an excessive compaction force may cause blockage of internal permeable voids. The adverse influence of the cement content on the coefficient of permeability (K) and porosity (P) may render the pervious functionality of PCBM inadequate and, hence, should not be ignored. Therefore, obtaining the optimal ranges of cement content and compaction force (corresponding to the on-site compaction force) through the proposed method is of great theoretical and practical significance for balancing the mechanical strength and coefficient of permeability of PCBMs.

## Implementation process and result analysis of the KS-ANN model

To achieve high predictive accuracy for the KS-ANN model, we first constructed statistical distribution models of the experimental data and used the Markov chain-Monte Carlo (MCMC) method to expand the experimental data. Based on this, a kriging-based surrogate (KS) model was executed to obtain results for the unconfined compressive strength (σ), porosity (P), and coefficient of permeability (K) under more combinations of working conditions. This mathematically expanded the experimental database while ensuring data continuity,

and the mean and variance values of the expanded database remained consistent with those of the previous database. Finally, the results of the KS model were used to train an ANN model, and the KS-ANN model was compared with the traditional ANN model.

## Analysis of Markov Chain Monte Carlo (MCMC) simulation results

The unconfined compressive strength (σ), porosity (P), and coefficient of permeability (K) obtained from laboratory tests under different testing conditions typically exhibit certain degrees of variability. For an ANN model, using test data with variable inputs helps it learn the discrete distribution characteristics of true data, thereby yielding predictive results that are closely aligned with actual conditions. Prior to MCMC simulations, the distribution model, mean, and variance of the data under different operating conditions need to be obtained. However, the data samples under each individual condition are limited, making it difficult to fit reliable statistical distribution models. Therefore, this study assumed that the data distribution under individual conditions follows the data distribution under group conditions. By analyzing the statistical distribution models of all the data, the data of the unconfined compressive strength (σ), porosity (P), and coefficient of permeability (K) were determined to follow a normal distribution. Therefore, a normal distribution model was used to fit the data of individual conditions to obtain their mean and variance values. Figure 6d displays the spatial distribution characteristics of the mean and variance of the unconfined compressive strength of the samples under different conditions (see below). The results serve as input for MCMC simulations for generating a massive amount of virtual data with the same distribution characteristics, which were used to train the KS model. The same processing method was employed for the porosity (P) and coefficient of permeability (K). Furthermore, as suggested by the ensemble learning concept[1,55], multiple random numbers can be used in MCMC to generate multiple initial datasets with the same mathematical distribution but different data. This allows for the training of multiple parallel KS-ANN models. The selection of random numbers can be customized based on the specific data requirements. Additionally, the predictions from these models can be averaged or voted to obtain results that meet the desired criteria. Therefore, in this paper, multiple random numbers were specified in the MCMC simulations to train multiple parallel KS-ANN models for predicting the unconfined compressive strength (σ), porosity (P), and coefficient of permeability(K), respectively.

Figure 2 presents the mean and variance values of the unconfined compressive strength (σ), porosity (P), and coefficient of permeability (K). The figure shows that the MCMC simulation results incorporate the statistical distribution characteristics of different physical indicators. By using the MCMC simulation results to train a kriging-based surrogate model, a kriging-based surrogate

### Table 2 | Design parameters of the cement paste in PCBM

| Cement content (%) | Cement (kg) | Water (kg) |
|---|---|---|
| 5 | 0.212 | 0.085 |
| 10 | 0.450 | 0.180 |
| 20 | 1.000 | 0.400 |

Where PCBM represents the permeable cement-stabilized base material.

### Table 3 | Laboratory test results for the uniaxial compressive strength, porosity, and coefficient of permeability of PCBMs

| Cement content (%) | Compaction force (kN) | σ (MPa) | P (%) | K (mm · s⁻¹) |
|---|---|---|---|---|
| 5 | 100 | 1.33/1.41/1.51 | 32.60/28.51/30.03 | 19.13/20.63/20.91 |
| 5 | 150 | 2.21/2.25/2.43 | 22.98/28.43/31.46 | 10.76/11.50/10.71 |
| 5 | 200 | 3.94/3.78/2.97 | 30.30/22.72/23.49 | 4.34/4.16/5.61 |
| 10 | 100 | 3.57/3.25/3.28 | 26.38/33.92/29.48 | 8.36/7.71/7.52 |
| 10 | 150 | 5.05/5.53/4.08 | 28.00/21.18/28.87 | 3.50/3.64/3.56 |
| 10 | 200 | 5.62/7.58/8.58 | 23.65/20.27/24.71 | 1.95/1.97/1.95 |
| 20 | 100 | 13.82/15.82/12.21 | 11.26/13.96/9.01 | 0.22/0.38/0.25 |
| 20 | 150 | 14.75/18.74/17.55 | 11.32/9.12/11.60 | 0.03/0.05/0.04 |
| 20 | 200 | 13.95/24.98/18.91 | 7.52/8.62/11.99 | 0.013/0.009/0.012 |

Where PCBM represents the permeable cement-stabilized base material, σ represents the unconfined compressive strength, P represents the porosity and K represents the coefficient of permeability.

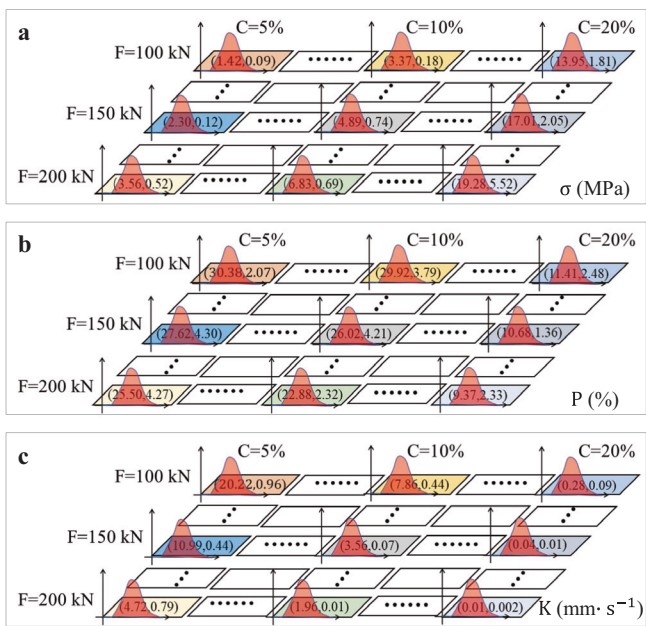

**Fig. 2 | Mean and variance results of the samples used for the Markov chain Monte Carlo (MCMC) simulations. a** Unconfined compressive strength (σ). **b** Porosity (P). **c** Coefficient of permeability (K).

model with desired statistical distribution characteristics is obtained accordingly. The results from the kriging-based surrogate model are ultimately used to train an ANN model. Therefore, the predicted results of the trained ANN model exhibit the discrete distribution characteristics of different physical indicators, which are consistent with the statistical characteristics reflected in the laboratory results.

In MCMC simulations, for each individual condition, 60,000 discrete data points are simulated, and the first 10,000 data points are discarded due to poor convergence. In this study, nine different combinations of conditions were considered, resulting in the simulation of 450,000 data points. Each condition included three physical indicators, resulting in the simulation of a total of 1,350,000 data points. Figure 3a–i present the MCMC simulation results. To make the labels in the figures clearly visible, we randomly selected a small number of MCMC generated data points for plotting. It is worth noting that the randomly selected data points represent the spatial distribution characteristics of all the data. As shown in Fig. 3a, the 5-100 (denoting cement content of 5% and compaction force of 100 kN) sample exhibits relatively small variability in terms of the unconfined compressive strength (σ), porosity (P), and coefficient of permeability (K). For samples 5-150 and 5-200 (see Fig. 3b, c), the variability in the spatial distribution of porosity (P) is significantly greater than that of the coefficient of permeability (K) or unconfined compressive strength (σ), with a data distribution that appears elongated. Furthermore, the samples with cement content of 10% (i.e., 10-100, 10-150, and 10-200, as shown in Fig. 3d–f) exhibit the same data distribution characteristics. However, for the samples with cement content of 20% (i.e., 20-100, 20-150, and 20-200, as shown in Fig. 3g–i), the variability in the spatial distribution of the unconfined compressive strength (σ) under different compaction forces is significantly greater than that of the porosity (P) or coefficient of permeability (K), with more significant differences observed for samples 20-150 and 20-200.

### Prediction results of the KS model
By using the proposed KS model, training was performed with the unconfined compressive strength (σ), porosity (P), and coefficient

of permeability (K) data by inputting parameter $x_i$, thereby obtaining the results shown in Fig. 3j–l. The figure shows that the spatial distribution pattern of the unconfined compressive strength (σ) (Fig. 3j) is opposite to that of the porosity (P) (Fig. 3k) or coefficient of permeability (K) (Fig. 3l), with porosity exhibiting the highest degree of variability. Specifically, as the unconfined compressive strength (σ) increases, the porosity (P) and permeability coefficient (K) decrease. Finally, the KS model data were standardized and input into an ANN model for training, which resulted in reliable weight files that can be used to predict the unconfined compressive strength (σ), porosity (P), and coefficient of permeability (K) of PCBMs under different conditions (or in-situ compacted conditions).

### Prediction results of the KS-ANN models
For a KS-ANN model, the greatest challenge is to determine the values of the parameters in the hidden layers. Herein, the trial and error strategy was used to continuously adjust the parameters in the hidden layers to minimize the prediction error of the KS-ANN model. After fine-tuning these parameters, the number of hidden layers in the ANN model was determined to be 3, with each hidden layer containing 128 neurons.

To evaluate the robustness of the KS-ANN model, a test dataset was created using the mean values of various physical indicators output by the KS model (see Fig. 4b). Additionally, another testing dataset was generated using discrete data of different physical indicators output by the KS model (see Fig. 4a). These testing datasets were used to verify the performance of the KS-ANN model. The results in Fig. 4b demonstrate that the KS-ANN model accurately predicts the mean of the data output based on the KS model, with a coefficient of determination ($R^2$) of 0.99. These findings confirm the reliability of the unbiased estimation of the KS model in surrogate modeling. Moreover, the results in Fig. 4a indicate that the KS-ANN model leads to an $R^2$ value of 0.94 when predicting the discrete distribution of data, further supporting the robustness of the KS-ANN model.

Figure 4c–e present the individual prediction results of the unconfined compressive strength (σ), porosity (P), and coefficient of permeability (K), respectively. Figure 4d shows that the porosity (P) has the lowest coefficient of determination (0.88), while the coefficient of permeability (K) has a coefficient of determination of 0.99 (Fig. 4e), exhibiting the best prediction result. The coefficient of determination of the unconfined compressive strength (σ) is somewhere in between (0.93) (Fig. 4c). In PCBMs, the unconfined compressive strength (σ) and coefficient of permeability (K) are direct indicators for evaluating whether or not the material meets the design requirements, and they can be accurately predicted by the well-trained KS-ANN model to optimize the design and compaction scheme.

To further verify the robustness of the KS-ANN model, we randomly selected 30% of the data obtained from laboratory tests as a supplementary test dataset to verify the prediction accuracy of the KS-ANN model. Note that this portion of data was intentionally excluded for use in the training datasets of any ANN model, i.e., it was solely used as an additional verification dataset. Figure 4g, h show the prediction results obtained by both the traditional ANN model and the proposed KS-ANN model for such a portion of laboratory test data. It can be seen from Fig. 4g that the coefficient of determination for the traditional ANN model is only 0.76; on the other hand, the proposed KS-ANN model leads to a coefficient of determination of 0.92 (see Fig. 4h), thus indicating a prediction accuracy improvement of 21% as compared to that of the traditional ANN model. This confirms that the use of the KS algorithm in the ANN model significantly improves the prediction accuracy. The proposed method makes it possible to apply machine learning in small-size databases, which has the potential to reduce experimental costs and enhance material design optimization.

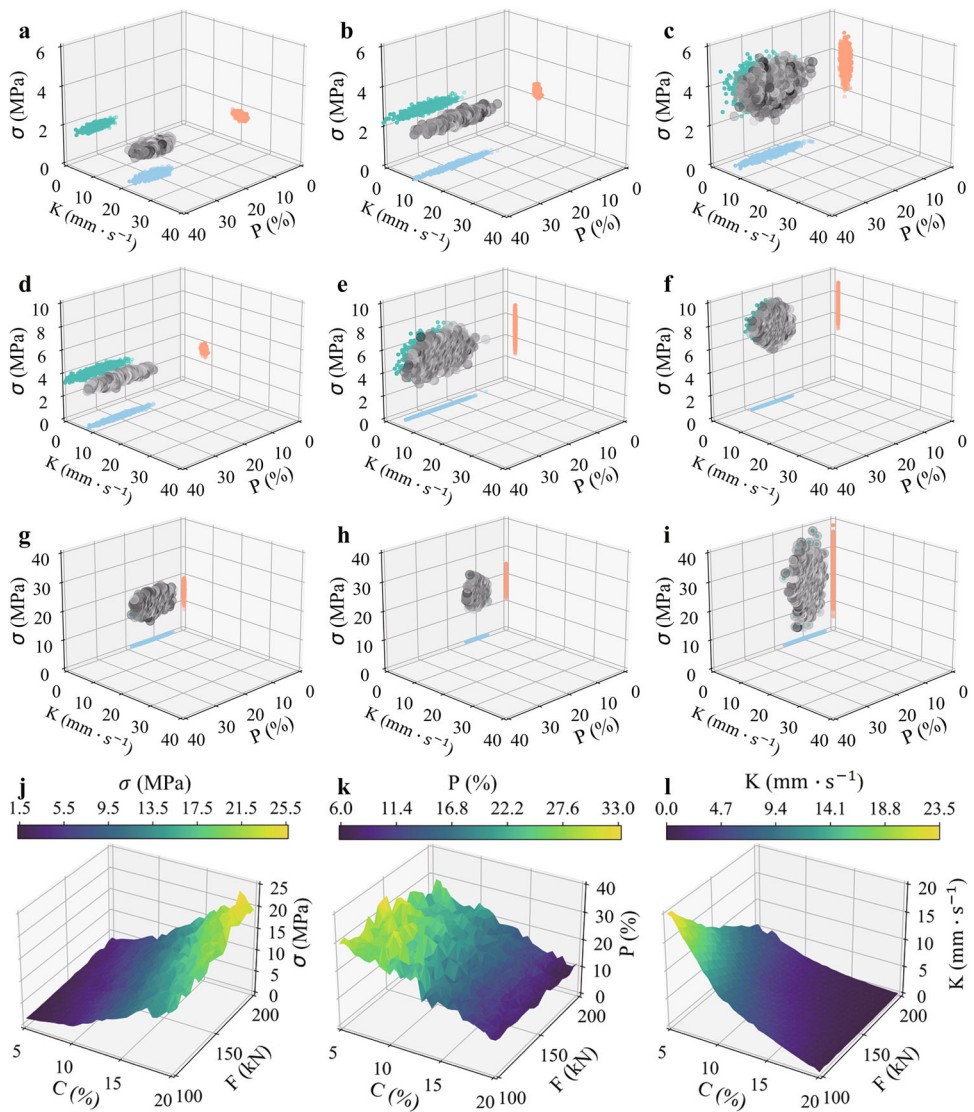

**Fig. 3 | Markov Chain Monte Carlo (MCMC) simulation and Kriging-based surrogate (KS) model prediction results.** MCMC simulation results of samples (**a** 5-100. **b** 5-150. **c** 5-200. **d** 10-100. **e** 10-150. **f** 10-200. **g** 20-100. **h** 20-150. **i** 20-200). Prediction results of the KS model for samples (**j** Unconfined compressive strength (σ). **k** Porosity (P). **l** Coefficient of permeability (K)).

## Data-driven design optimization concept of the PCBM

A balance between the unconfined compressive strength and coefficient of permeability is a prerequisite for the widespread applications of PCBMs. Although the design method stemming from cement-stabilized dense-graded base materials may improve the strength of PCBMs, their coefficient of permeability may not meet the requirements. Therefore, when either laboratory sample preparation or on-site compaction is employed, PCBMs should simultaneously meet the requirements of the unconfined compressive strength and coefficient of permeability. Additionally, design optimization through methods such as orthogonal design in laboratory sample preparation or on-site compaction is time-consuming and costly. To address these issues, the proposed method predicts the unconfined compressive strength and coefficient of permeability under different working conditions, including sample preparation conditions that cannot be achieved in laboratory tests. Based on this, a design optimization method is proposed for PCBMs with the unconfined compressive strength and coefficient of permeability targeted. The specific implementation procedure is described as follows:

(1) Based on the correspondence between laboratory sample preparation and field compaction conditions, the random variables $x_i = [C_i, F_i]$ are determined, which are directly related to the unconfined compressive strength and coefficient of permeability. A small number of orthogonal experiments are designed to obtain the physical indicators $Y_i$ [i.e., the unconfined compressive strength (σ), porosity (P), and coefficient of permeability (K)] under different random variables $x_i$, which serve as input parameters for the KS-ANN model.

(2) To obtain the optimal sample preparation or compaction scheme, a performance function is established for the optimization scheme. Typically, the unconfined compressive strength (σ) and coefficient of permeability (K) are selected as performance evaluation indicators of laboratory sample preparation or field compaction results. The expression of the performance function is as follows:

$$\begin{cases} \sigma(\boldsymbol{x_i}) \geq \sigma_{\min} \\ K(\boldsymbol{x_i}) \geq K_{\min} \end{cases} \quad (1)$$

where $\sigma_{\min}$ and $K_{\min}$ are the minimum values specified by related specifications or regulations[54], which are preselected as 3.5 MPa and 0.5 mm · s$^{-1}$ in this study, respectively.

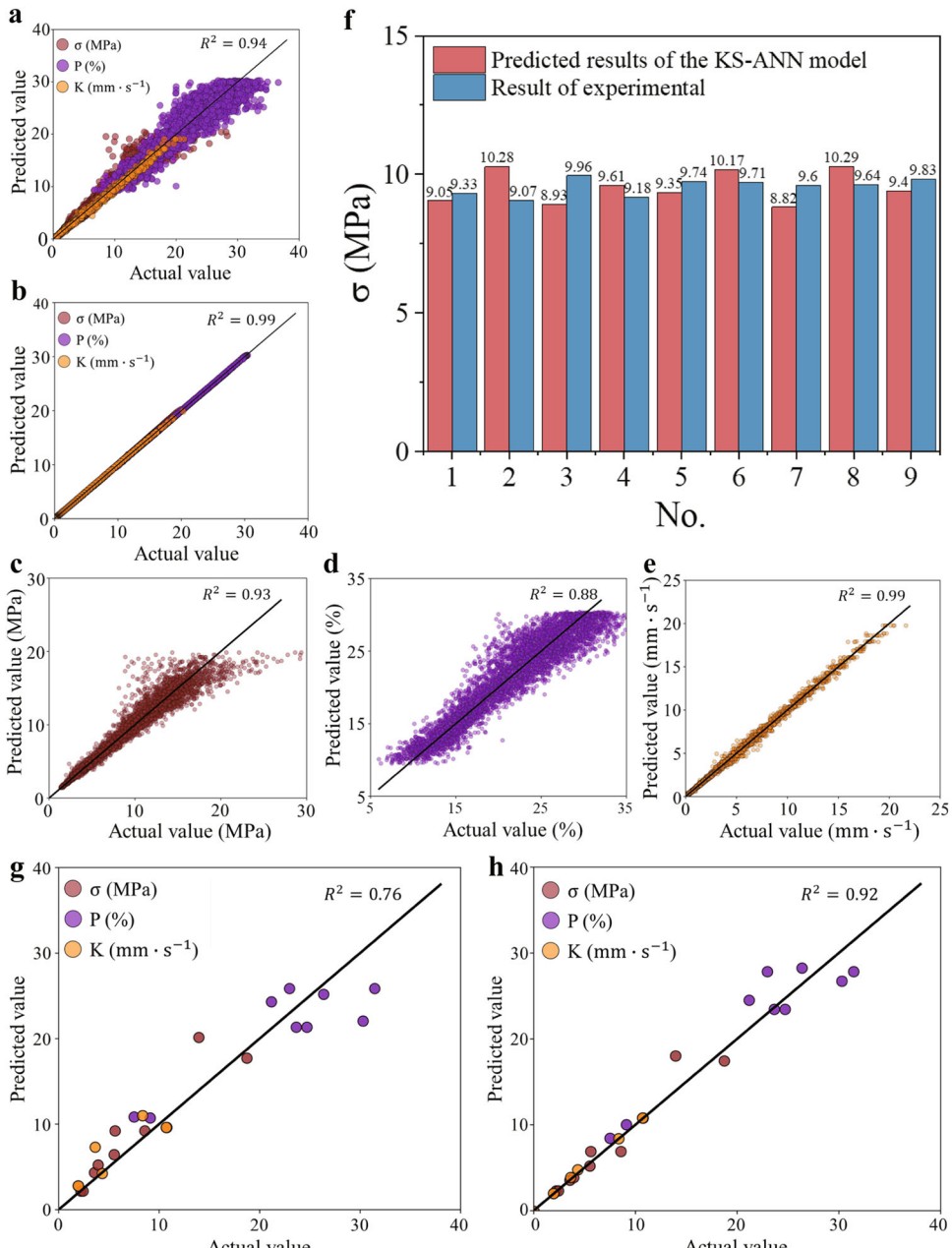

**Fig. 4 | Prediction results of the ANN/KS-ANN model.** Prediction results of the KS-ANN model (**a**) Test dataset of different physical indicators output by the KS model. **b** Test dataset of mean values of different physical indicators output by the KS model. **c** Test dataset of the unconfined compressive strength (σ) output by the KS model. **d** Test dataset of porosity (P) output by the KS model. **e** Test dataset of the coefficient of permeability (K) output by the KS model. **f** Comparison between the results predicted by the KS-ANN model and the experimental results of the unconfined compressive strength (the specimens were prepared with a cement content of 10% and subjected to a compaction force of 300 kN). **g** Prediction results of the ANN on the test dataset from the experimental data. **h** Prediction results of the KS-ANN on the test dataset from the experiment. Note that ANN represents the artificial neural network, and KS-ANN represents the kriging-based surrogate data-enriching artificial neural network.

## Design optimization function for PCBMs

Figure 5a–b illustrate the prediction results of the KS-ANN model for the unconfined compressive strength (σ) (Fig. 5a) and coefficient of permeability (K) (Fig. 5b) over a wide range, respectively. As shown in the figure, the distribution characteristics of the unconfined compressive strength (σ) and coefficient of permeability (K) are consistent with the patterns observed in laboratory tests. To obtain an optimized function, data points meeting the performance function over a wide range need to be obtained. Therefore, the optimization function aims to find the contour that meets the performance function in the matrix plot of Fig. 5a–b, which is the desired design optimization function.

In order to verify the predictive performance of the KS-ANN model against external datasets beyond the training datasets used, 9 replicate specimens were prepared in the laboratory with a cement content of 10% and a compaction force of 300 kN for conducting unconfined compression tests. Figure 4f shows the comparison between the unconfined compressive strength (σ) results predicted by the KS-ANN model and the experimental measurements. Consistent with the experimental results, the KS-ANN model utilized nine different random numbers to predict the unconfined compressive strength (σ). It is evident from Fig. 4f that the predicted results of the KS-ANN model exhibit dispersity characteristics that

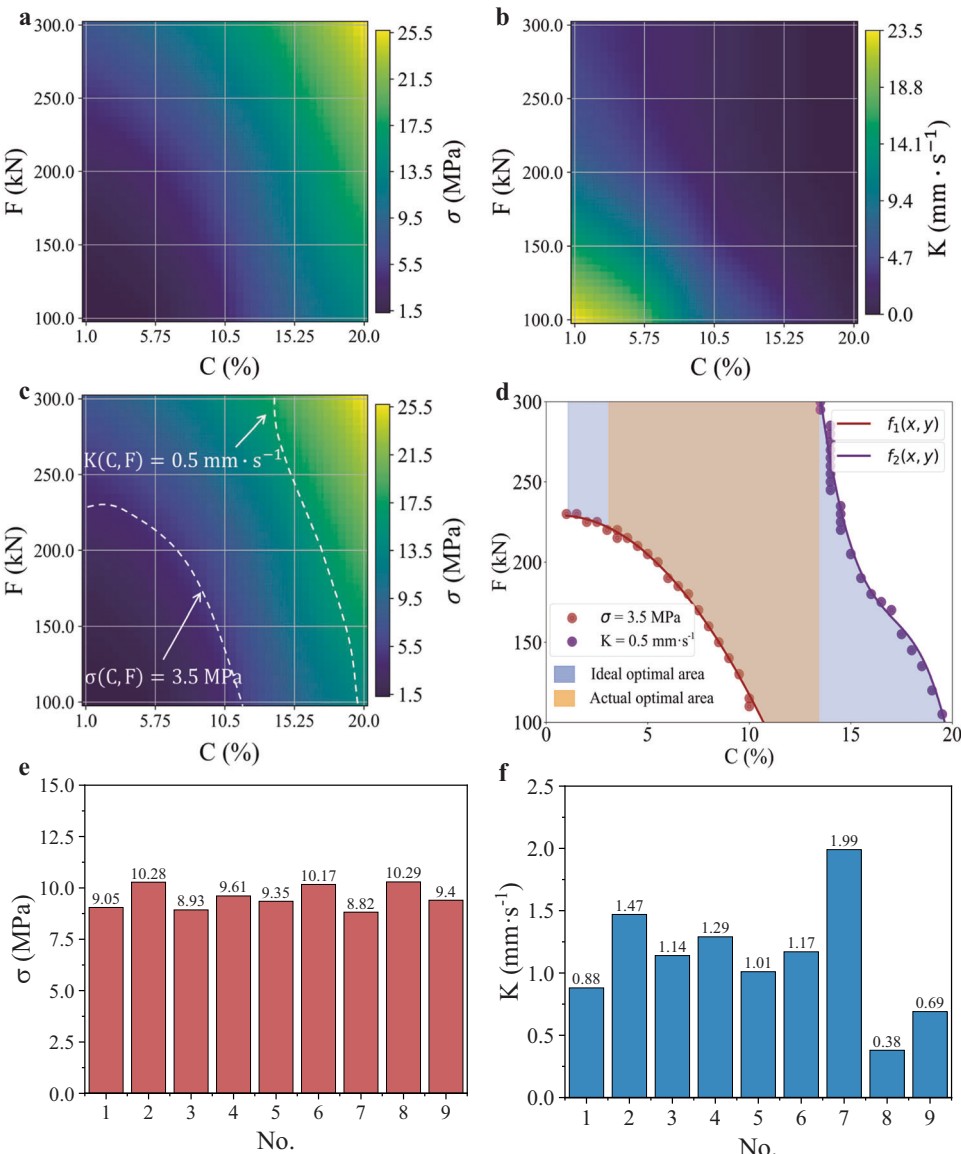

**Fig. 5 | Design optimization process of PCBM corresponding to actual engineering applications. a** Matrix plot of the unconfined compressive strength (σ) results predicted by the KS-ANN model. **b** Matrix plot of the coefficient of permeability (K) results predicted by the KS-ANN model. **c** Construction process of the design optimization function, where the dashed line corresponding to σ(C,F) = 3.5MPa represents the contour line of unconfined compressive strength

equal to 3.5 MPa and the dashed line corresponding to K(C,F) = 0.5mm · s⁻¹ represents the contour line of coefficient of permeability equal to 0.5mm · s⁻¹. **d** Results of solving the design optimization function. **e** Results predicted by the KS-ANN model for unconfined compressive strength (σ). **f** Results predicted by the KS-ANN model for the coefficient of permeability (K). Note that F represents the compaction force and C represents the cement content.

are similar in magnitude and trend to the laboratory test results. The average value of the results predicted by the KS-ANN model was approximately 9.54 MPa, while the average value of the laboratory test results was approximately 9.56 MPa. The difference was only 0.02 MPa. Therefore, this finding verifies the robustness of the KS-ANN model.

Figure 5c, d display both the construction and solving processes of the design optimization function. First, a self-developed Python® program is used to solve the points satisfying Eq. (1) on the matrix graph of the predicted results of the unconfined compressive strength (σ) and coefficient of permeability (K). Then, the equivalent points are projected onto the same matrix graph (Fig. 5c). The obtained equivalent points are retained on the matrix graph, which is converted into the regular Cartesian coordinate system. By fitting the equivalent points, the desired design optimization function can be obtained (Fig. 5d). According to Fig. 5d, the ideal optimal region is

bounded by the optimization function and the coordinate axes. For instance, when the compaction force is 300 kN, the maximum acceptable cement content is 13.5%. However, in actual engineering practice, the cement content of PCBMs is typically not less than 3%. Besides, actual PCBMs need to achieve a suitable degree of compaction, that is, a compaction force of 300 kN is often needed. Therefore, the optimal cement content range that can simultaneously meet the requirements of the unconfined compressive strength (σ) and coefficient of permeability (K) is between 3% and 13.5%.

Through the above process, an optimization function denoted by Eq. (2) was formulated. Given the specified cement content, it becomes feasible to anticipate the range of compaction force that fulfills both the unconfined compressive strength and coefficient of permeability simultaneously. It is important to bear in mind that during the practical compaction process of PCBMs, the minimum compacting force, $F_{min}$,

generally exceeds 100 kN, while the maximum value, $F_{max}$, typically does not exceed 300 kN.

$$\begin{cases} F_{min} = 228.61 + 1.57C - 1.27C^2 \quad R^2 = 0.99 \\ F_{max} = 10045.3 - 1706.09C + 98.95C^2 - 1.93C^3 \quad R^2 = 0.96 \end{cases} \quad (2)$$

where $F_{min}$ and $F_{max}$ are the minimum and maximum compaction forces (in unit of kN), respectively; and $C$ is the designed cement content (in unit of %).

After obtaining the optimization functions of PCBM, with the assumption that the cement content of PCBM is designed to be 10% in related engineering practices, the range of compaction force demanded by Eq. (1) can be calculated as [117.31, 300.00] according to Eq. (2). This indicates that the minimum compaction force should not be lower than 117.31 kN, while the upper limit of the compaction force is determined to be 300 kN by the on-site construction process. Moreover, laboratory specimens with a cement content of 10% were prepared using a compaction force of 300 kN to validate the effectiveness of the optimization functions. Since the unconfined compressive strength results in Fig. 4f confirm the validity and credibility of the developed KS-ANN model, the KS-ANN model is then utilized to predict the unconfined compressive strength and coefficient of permeability of laboratory specimens with a cement content of 10% and a compaction force of 300 kN. As shown in Fig. 5e, f, the mean value of the unconfined compressive strength predicted by the KS-ANN model is approximately 9.54 MPa, whereas the mean value of the coefficient of permeability is approximately 1.11 mm $\cdot$ s$^{-1}$. These values satisfy the functionality and performance requirements. The above results confirm the effectiveness of the developed optimization functions and corresponding strategy, which could guide the design optimization and compaction scheme determination for PCBMs.

In this study, a new KS-ANN model is proposed and its robustness is verified. The well-trained KS-ANN model was then used to predict the unconfined compressive strength ($\sigma$) and coefficient of permeability (K) of PCBMs over a wide range. This study provides technical references for material design optimization and laboratory or in-situ compaction scheme determination. The main findings are as follows.

(1) The proposed method effectively integrates real laboratory tests, mathematical modeling, and ML methods, making full use of all available information (which is often limited in reality). Compared to traditional ANN models, the proposed KS-ANN model improved the prediction accuracy by 21% for samples with small datasets.

(2) The trained KS-ANN model exhibited high accuracy as indicated by the coefficient of determination (R$^2$) of 0.99 while predicting the mean of the KS model, as well as by the average accuracy (R$^2$ = 0.94) while predicting the discrete distribution features of the unconfined compressive strength ($\sigma$), porosity (P), and coefficient of permeability (K). The trained KS-ANN model can be used to accurately predict the macroscopic hydromechanical properties of PCBMs, allowing for cost-effective optimization of its design and compaction strategies.

(3) Based on the prediction results of the KS-ANN model, an optimized function that can simultaneously satisfy the requirements of the unconfined compressive strength ($\sigma$) and coefficient of permeability (K) was obtained, which can accurately calculate the optimal cement content and compaction force ranges of PCBMs. The proposed method can significantly reduce the experiment cost and provide a heuristic theoretical reference for predicting key hydromechanical properties, design optimization, and engineering practices for similar materials.

(4) The proposed method uses the data-enriching technique assisted by a kriging-based surrogate model, which is highly beneficial for enhancing the predictive accuracy and efficiency of the ANN

model. However, to extend the use of this proposed method in other engineering applications, the input data must satisfy the requirements of the kriging-based surrogate models, i.e., they cannot exceed 4 dimensions. This is the major limitation of this proposed method, which is being tackled in the research efforts currently underway.

## Methods

### Framework of the KS-ANN model

Kriging-based surrogate (KS) models advantageously afford highly accurate predictions within a domain, even with a limited number of samples. However, according to the previous studies by Zeng et al.[14] and Sun et al.[15], the prediction results of the KS model may not converge outside the domain of the real data or outside the boundary. Therefore, real data from the entire domain need to be obtained when practically applying KS models, which can be challenging under strict laboratory testing conditions. In contrast, ANN models not only exhibit satisfactory prediction accuracy within specific domains but also yield trustworthy prediction results when applied to domains reasonably beyond their initial training set. However, disadvantageously, ANN models require substantial amounts of sample data to form the training set, which inevitably results in high costs and requires significant effort and time. By combining the advantages of KS models and ANN models, accurate data prediction across the entire domain can be realized with a small number of laboratory test samples, thereby compensating for the deficiencies of laboratory tests and reducing the cost.

The proposed KS-ANN framework for obtaining reliable prediction accuracy with a limited number of data samples is shown in Fig. 6. First, laboratory tests are conducted to observe the hydromechanical performance parameters of the PCBM under different combinations of testing conditions (Fig. 6a–c). The proposed combination of testing conditions considered herein includes the cement content of the sample and the static compaction force during the sample preparation, corresponding to the load applied by the roller compactor in the field construction sites. The obtained hydromechanical performance parameters are the unconfined compressive strength ($\sigma$), porosity (P), and coefficient of permeability (K). Then, the statistical distribution models of each hydromechanical performance parameter under the specific operating conditions are constructed to replace their actual observation results (Fig. 6d), and the Markov chain Monte Carlo (MCMC) method is used to simulate abundant data samples, which completely agree with the statistical distribution models and their metrics of each hydromechanical performance parameter (Fig. 6e). Subsequently, the simulation results are used to train a reliable KS model. In the KS model, numerous virtual combinations of working conditions are interpolated, and the trained KS model is used to predict hydromechanical performance parameters under different combinational conditions (Fig. 6f). Subsequently, a large number of high-precision virtual samples distributed within the domain are obtained. Finally, the obtained output data of the KS model are used as a training dataset for the ANN model to train a reliable weight model for predicting the hydromechanical performance indicators of the PCBM, including the uniaxial compressive strength ($\sigma$), porosity (P), and coefficient of permeability (K). Hence, it is necessary to construct a multi-output artificial neural network (ANN) model to meet the prediction needs of PCBMs (Fig. 6g). Additional details can be found in related work[48].

### Kriging-based surrogate (KS) model

A substantial number of data samples is a fundamental prerequisite for successful ANN model training. However, obtaining a significant amount of data may involve tens of thousands of repetitive laboratory tests, which is challenging in practical engineering applications.

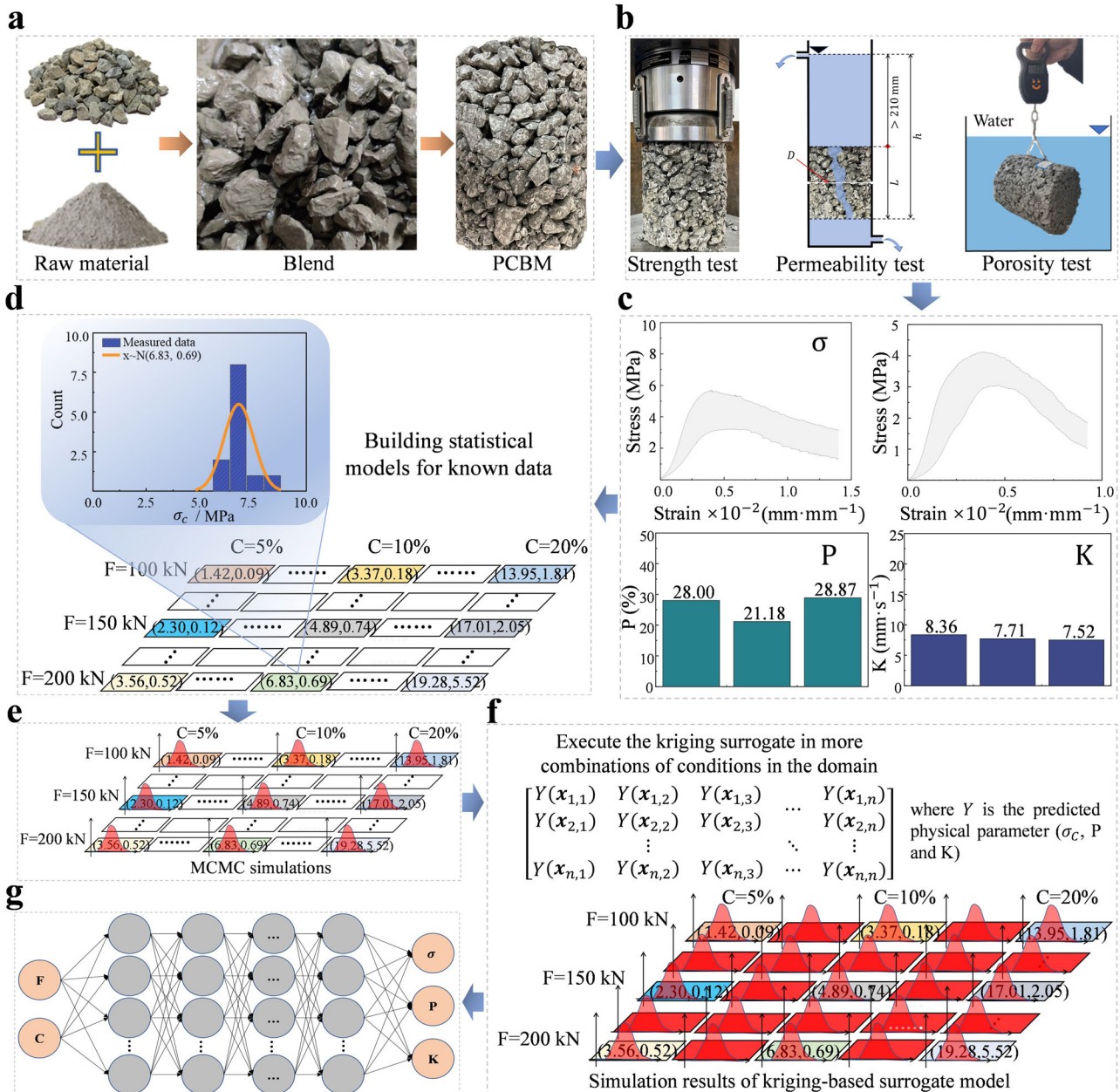

**Fig. 6 | Proposed framework for using a kriging-based surrogate data-enriching artificial neural network (KS-ANN) for improving the prediction accuracy of the strength and coefficient of permeability of PCBM. a** Preparation of PCBM samples. **b** Laboratory testing. **c** Obtain the unconfined compressive strength (σ), porosity (P), and coefficient of permeability (K) of PCBM. **d** Building a statistical distribution model for known data, where the focus of the color shading marker represents the statistical distribution of hydromechanical performance indicators under the combination of C and F. **e** Construction of Markov chain Monte Carlo (MCMC) simulations based on statistical distribution models of known data. **f** Enriching data within a domain using KS models. **g** Network structure of the ANN model. Note that F represents the compaction force, C represents the cement content, PCBM represents the permeable cement-stabilized base material, and MCMC represents the markov chain monte carlo.

Therefore, this study proposes the use of KS models to train the relationship between random input variables $x_i = [C_i, F_i]$ and the hydromechanical performance indicators ($Y$) [i.e., the unconfined compressive strength (σ), porosity (P), and coefficient of permeability (K)] under different operating conditions. A trained KS model can be used to effectively predict the hydromechanical performance indicators under different laboratory test conditions within the domain.

KS models are semiparametric surrogate models based on statistical theory[18,56–58]. They accurately predict data samples within a domain, thereby providing additional predicted samples within the domain that have the same physical significance as real samples.

Typically, the response function of KS models comprises parametric linear regression and nonparametric stochastic processes[14,15,59,60]:

$$\hat{\Gamma}(x) = f(x)^T \beta + g(x) \tag{3}$$

where $\hat{\Gamma}(x)$ is the predicted value of a kriging-based model for the response function $\Gamma(x)$. $f(x) = [f_1(x), f_2(x), f_3(x), \ldots f_k(x)]^T$ is a vector of basic functions. $\beta = [\beta_1, \beta_2, \beta_3, \ldots \beta_n]$ is a vector of trend coefficients. $g(x)$ is a random Gaussian process with a mean of 0 and a variance of $\delta^2$, where $\delta^2$ is the variance of the input data. The unbiased estimator $(E(\hat{\Gamma}(x) - \Gamma(x)) = 0)$ guarantees the stability of the random Gaussian process. Here, the covariance of any two points $x_i$ and $x_j$ can be

calculated as follows:

$$Cov[g(\boldsymbol{x}_i), g(\boldsymbol{x}_j)] = \delta^2 R(\boldsymbol{\theta}, \boldsymbol{x}_i, \boldsymbol{x}_j) \qquad (4)$$

where $R(\boldsymbol{\theta}, \boldsymbol{x}_i, \boldsymbol{x}_j)$ represents the correlation function and θ is a pre-defined model parameter. Typically, the correlation function includes linear, spherical, and Gaussian models. The Gaussian model employed in this study is defined as:

$$R\left(\boldsymbol{\theta}, \boldsymbol{x}_i, \boldsymbol{x}_j\right) = \prod_{p=1}^{m} \exp\left(-\theta_p d_p^2\right), p = 1,2,3,\ldots,m \qquad (5)$$

where $\theta_p$ represents the p-th component of $\boldsymbol{\theta}$. $d_p = x_{ip} - x_{jp}(p = 1,2,3,\ldots,m)$ is the distance between the p-th coordinates of the sample points $\boldsymbol{x}_i$ and $\boldsymbol{x}_j$. $m$ represents the dimensionality of the coordinates.

Consider a dataset for training denoted as $\boldsymbol{x} = [\boldsymbol{x}_1, \boldsymbol{x}_2, \boldsymbol{x}_3, \ldots, \boldsymbol{x}_n]^T, (\boldsymbol{x}_i = [x_{i1}, x_{i1}, x_{i1}, \ldots, x_{i1}] i = 1,2,3,\ldots,n)$ and a performance response function denoted as $\boldsymbol{\Gamma} = [\Gamma(\boldsymbol{x}_1), \Gamma(\boldsymbol{x}_2), \Gamma(\boldsymbol{x}_3), \ldots, \Gamma(\boldsymbol{x}_N)]^T$. Then, the maximum likelihood estimation for $\boldsymbol{\beta}$ and $\delta^2$ can be obtained as follows:

$$\begin{cases} \hat{\delta}^2 = \frac{1}{m}\left(\boldsymbol{\Gamma} - \mathbf{F}\hat{\boldsymbol{\beta}}\right)^T \boldsymbol{R}^{-1}(\boldsymbol{Y} - \mathbf{F}\hat{\boldsymbol{\beta}}) \\ \hat{\boldsymbol{\beta}} = (\boldsymbol{F}^T \boldsymbol{R}^{-1} \boldsymbol{F})^{-1} \boldsymbol{F}^T \boldsymbol{R}^{-1} \boldsymbol{\Gamma} \end{cases} \qquad (6)$$

where $\mathbf{F} = [\boldsymbol{f}^T(\boldsymbol{x}_1), \boldsymbol{f}^T(\boldsymbol{x}_2), \boldsymbol{f}^T(\boldsymbol{x}_3), \ldots, \boldsymbol{f}^T(\boldsymbol{x}_N)]^T$ is a regression coefficient matrix. $R_{ij} = R(\boldsymbol{\theta}, \boldsymbol{x}_i, \boldsymbol{x}_j)(i, j = 1,2,3,\ldots, N)$ is a correlation matrix. $\boldsymbol{Y}$ is the target value of dataset $\boldsymbol{x}$. The parameter $\boldsymbol{\theta}$ of the model can be obtained as follows:

$$\boldsymbol{\theta} = \arg\min|\boldsymbol{R}|^{1/n}\hat{\delta}^2 \qquad (7)$$

In KS models, for an unknown point $\boldsymbol{x}^u$, the predicted value of $\hat{\Gamma}(\boldsymbol{x}^u)$ can be calculated using a Gaussian random process, wherein $\hat{\Gamma}(\boldsymbol{x}^u)$ follows a Gaussian distribution with mean $\mu_{\hat{\Gamma}}(\boldsymbol{x}^u)$ and variance $\hat{\delta}_{\hat{\Gamma}}^2(\boldsymbol{x}^u)$. This can be expressed as $\hat{\Gamma}(\boldsymbol{x}^u) \sim N(\mu_{\hat{\Gamma}}(\boldsymbol{x}^u), \hat{\delta}_{\hat{\Gamma}}^2(\boldsymbol{x}^u))$. The predicted mean value $\mu_{\hat{\Gamma}}(\boldsymbol{x}^u)$ and the predicted variance $\hat{\delta}_{\hat{\Gamma}}^2(\boldsymbol{x}^u)$ are given as

$$\begin{cases} \mu_{\hat{\Gamma}}(\boldsymbol{x}^u) = \boldsymbol{f}^T(\boldsymbol{x}^u)\hat{\boldsymbol{\beta}} + \boldsymbol{r}^T(\boldsymbol{x}^u)\boldsymbol{R}^{-1}(\boldsymbol{Y} - \mathbf{F}\hat{\boldsymbol{\beta}}) \\ \hat{\delta}_{\hat{\Gamma}}^2(\boldsymbol{x}^u) = \hat{\delta}^2 - \left[\boldsymbol{f}^T(\boldsymbol{x}^u)\boldsymbol{r}^T(\boldsymbol{x}^u)\boldsymbol{R}^{-1}\right]\begin{pmatrix} \mathbf{0} & \boldsymbol{F}^T \\ \boldsymbol{F} & \boldsymbol{R} \end{pmatrix}^{-1}\begin{bmatrix} \boldsymbol{f}(\boldsymbol{x}^u) \\ \boldsymbol{r}((\boldsymbol{x}^u) \end{bmatrix} \end{cases} \qquad (8)$$

where $\boldsymbol{r}^T(\boldsymbol{x}^u) = [\boldsymbol{R}(\boldsymbol{\theta}, \boldsymbol{x}^u, \boldsymbol{x}_1), \boldsymbol{R}(\boldsymbol{\theta}, \boldsymbol{x}^u, \boldsymbol{x}_2), \boldsymbol{R}(\boldsymbol{\theta}, \boldsymbol{x}^u, \boldsymbol{x}_3), \ldots, \boldsymbol{R}(\boldsymbol{\theta}, \boldsymbol{x}^u, \boldsymbol{x}_N)]$ is the correlation vector between input training points.

KS models aim to construct the most accurate surrogate model with a small amount of training data. Therefore, they require an appropriate sampling strategy[61]. The mixed adaptive sampling strategy is widely used due to its advantages of spatial filling and adaptability[11]. The hybrid adaptive sampling method selects a new sampling point by maximizing the quality parameter $s_j$ at each candidate sampling point $\boldsymbol{x}_j(j = 1,2,3,\ldots k$, where $k$ is the number of candidate sample points. $s_j$ is calculated as follows:

$$s_j = \frac{d_j}{\max(d_j)} + \frac{\hat{\delta}_j^2}{\max(\hat{\delta}_j^2)} \qquad (9)$$

where $d_j$ represents the minimum Euclidean distance between the $j$-th candidate sample point and the current design sample point (including the initial and newly inserted sample points). $\hat{\delta}_j^2$ is the predicted

variance at the $j$-th candidate sample point that is directly provided by a KS model.

## Artificial Neural Network (ANN) Model

ANNs simulate the functionality of biological neurons to process various nonlinear information inputs and accurately predict output results after fully learning the input information characteristics, as shown in Fig. 1f. The construction of ANN models comprises three main steps: (1) defining the input and output of the information, (2) training the hidden and output layers of the ANN by calculating weights and continuously reducing errors, and (3) evaluating the ANN accuracy by comparing the predicted and actual values. In this study, the input and output layers are kept fixed. The input layer includes the static compaction force (F) and cement content (C), while the output layer includes the unconfined compressive strength (σ), porosity (P), and coefficient of permeability (K), representing a typical regression problem.

The accuracy of data prediction using an ANN model depends on the quality of the training set, but obtaining large amounts of data from laboratory tests is not economical. Therefore, this study proposes a KS model to reduce the number of experiments and obtain high-quality data with real physical significance for training the ANN model. The proposed learning rate for the ANN model is set as 0.002, the maximum number of iterations is 2000, the convergence error is 0.001, and the ratio of the training set to the test set is 9:1. Before commencing the training process, the input data $\boldsymbol{x}$ are standardized and the output data are kept unchanged, as is customary. The standardized dataset $\boldsymbol{x}^*$ has a mean value of 0 and a standard deviation of 1. The data are standardized using the following formula:

$$\boldsymbol{x}^* = \frac{\boldsymbol{x} - \mu}{\eta} \qquad (10)$$

where $\boldsymbol{x}^*$ and $\boldsymbol{x}$ represent the standardized and original input data, respectively. Additionally, $\mu$ and $\eta$ are the mean and variance of data $\boldsymbol{x}$, respectively.

Moreover, the mean squared error (MSE) between the predicted and actual values is calculated using Eq. (11). The proposed ANN model aims to minimize the MSE during the iterative process, thus achieving high predictive accuracy.

$$MSE = \frac{1}{N}\sum_{i=1}^{N}(Y_i^t - Y_i^p)^2 \qquad (11)$$

where N represents the quantity of data in the testing dataset. $Y_i^t$ and $Y_i^p$ are the actual and predicted values in the testing dataset, respectively.

To assess the accuracy of the proposed ANN model, the coefficient of determination ($R^2$) is employed to evaluate both the training and prediction datasets to prevent underfitting or overfitting of the ANN model. The coefficient of determination is defined as follows:

$$R^2 = \frac{\sum_{i=1}^{N}Y_i^p - \bar{Y}_i^p}{\sum_{i=1}^{N}Y_i^{actual} - \bar{Y}_i^{actual}} \qquad (12)$$

where $Y_i^{actual}$ represents the actual data of the testing dataset. $\bar{Y}_i^p$ and $\bar{Y}_i^{actual}$ represent the average predicted and actual values of the testing dataset, respectively. The constructed KS-ANN model can promptly and accurately predict the hydromechanical performance indicators of PCBMs under different sample preparation conditions across a wide range.

## Reporting summary

Further information on research design is available in the Nature Portfolio Reporting Summary linked to this article.

## Data availability

The data generated in this study are provided in the Supporting Information/Source Data file. Data files for model training and testing generated in this study have been deposited in the public GitHub (https://github.com/YM008/Supporting-Materials.git) without any restrictions. The data that support the plots within this paper and other findings of this study are available from the corresponding authors upon request. Source data are provided in this paper.

## Code availability

The Python code to implement the machine learning tasks in this study has been deposited in the public database Zenodo[62] without any restrictions.

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

## Acknowledgements

This work was jointly supported by the Science Fund for Distinguished Young Scholars of Hunan Province, China (2024JJ2073 to Y.X.), National Natural Science Foundation of China (52178443, 51878673, & 51808577 to Y.X.), the National Key R&D Program of China (2019YFC1904704 to Y. Chen, Z. Li, and Y.X.), and the Fundamental Research Funds for the Central Universities of Central South University, China (2023zzts0405, 2021zzts0227, 2021zzts0223 & 2022ZZTS0744 to X. Wang, M. Wang, and W. Li). The computing resources provided by the High-Performance Computing Center of Central South University are gratefully acknowledged.

## Author contributions

Xiaoming Wang: Conceptualization, Data curation, Formal analysis, Investigation, Methodology, Software, Validation, Visualization, Writing – original draft, Writing – review & editing. Yuanjie Xiao: Conceptualization, Data curation, Formal analysis, Funding acquisition, Investigation, Methodology, Project administration, Resources, Software, Supervision, Validation, Visualization, Writing – original draft, Writing–review & editing. Wenqi Li: Data curation, Formal analysis, Investigation, Validation, Writing – review & editing. Meng Wang: Data curation, Formal analysis, Investigation, Validation, Writing – review & editing. Yanbin Zhou: Data curation, Methodology, Visualization, Writing – review & editing. Yuliang Chen: Conceptualization, Investigation, Methodology, Supervision, Validation, Writing – review & editing. Zhiyong Li: Conceptualization, Investigation, Methodology, Supervision, Validation, Writing – review & editing.

## Competing interests

The authors declare no competing interests.
