## [Peer Review File · Nature Communications]

Kriging-based Surrogate Data-enriching Artificial Neural Network Prediction of Strength and Permeability of Permeable Cement-stabilized BaseREVIEWER COMMENTS

Reviewer #1 (Remarks to the Author):

This paper presents a combined method of real laboratory tests, mathematical modeling, and machine learning using a kriging-based surrogate model-aided artificial neural network (KS-ANN) to reduce laboratory testing costs and improve predictive accuracy. The proposed method was verified by limited test data of permeable cement-stabilized base materials (PCBMs). In general, this research gives a significant reference for rapid prediction of some required test parameters when the related sampling size is small due to limited test cost and time. Meanwhile, the manuscript is organized well with acceptable readability as well as originality and novelty. However, some following concerns should be discussed if considered for publication,

(1) The proposed KS-ANN framework is a general method or algorithm to do the accurate extending prediction for the small size of test sampling data. The prediction of PCBMs' behavior is just a case study for verification of this method. Therefore, the question is how to understand the focus of this research. For the current title, it is recommended to be revised to emphasize the general method while not the PCBMs.

(2) The mechanical strength and permeability of PCBMs are easy to test in practice, so another question is the necessity for the application of this proposed method. Please enhance the background to explain the reason for verifying the KS-ANN method using PCBMs or consider more appropriate applications like some complicated laboratory tests as the case recommendation.

(3) The basic idea of this research could be summarized in two steps. The first is that a limited number of lab test results were used to obtain the statistical distribution parameters of the input and output variables. Then, Monte Carlo sampling methods were applied to generate more sample data to train the ANN model. Here, the question is that the ANN model is trained by virtual data generated by the Monte Carlo method even though the limited laboratory data, as we know, any model is better trained by real data, so how to explain the reliability of the prediction by the trained ANN model in theoretical view?

(4) Some more background on the KS model could be provided to explain the conflict between highly accurate predictions and strong divergence indicated in Line 105 to Line 106. In addition, the Monte Carlo can generate samples based on the distribution function of data in general. What is the purpose of using Markov chain? Please clarify.

(5) For the case study of PCBMs, the real test data from three different static compaction forces, 100, 150, and 200 kN, and the compaction force of 300 kN is used for prediction. The case is okay, but some other cases in the output are nonlinear or complicated concerning the input in different regions may not be suitable. Please explain.

(6) It is better to compare the difference in accuracy between a model trained on the expanded dataset and a model trained on the experimental dataset. Moreover, validation should be performed on real data to verify the applicability of the model in the real world.

(7) Are variables related to working conditions also random variables with specific distributions? How do these conditions correspond to the physical parameters sampled from the distribution? How can it be verified that such a correspondence is consistent with the actual conditions?

(8) From this research, "a statistical distribution model is established using actual laboratory test data. It is used to generate input parameters for Markov Chain Monte Carlo (MCMC) simulations and a kriging-based surrogate model" If the lab test results can be used to obtain the statistical distribution, what is the purpose of using the KS model? Please clarify. It seems the ANN can be built without any input from the KS. The key is the MC sampling.

(9) Are the MCMC simulations of the three physical parameters performed separately? If so, how

was the correspondence between the simulation results determined, i.e., how were the 3D coordinates of each point in Figure 6 determined? The markers of the points in Figure 6 are difficult to distinguish and need to be modified.

(10) Compared to transfer learning, what are the significant advantages of KS-ANN? In addition, with the current rise of big model approaches to learning by migration, does virtual data based on agent models have the potential to serve as the underlying data for migration learning? Please supplement some more information into the literature review.

(11) Please explain the difference between Fig 8 (a) and (b). Meanwhile, some of the data labels in Figures were very hard to read. Please check and improve the quality of the charts.

Reviewer #2 (Remarks to the Author):

This paper aims to construct a Kriging-Based Surrogate Model combined with ANNs using experimental data to predict key properties of porous cement-based concrete materials. We strongly believe that this is the first instance of a data enrichment technique empowering a machine learning model applied to conventional pervious concrete. While it presents interesting aspects, there are certain unclear points that require revision, as outlined below:

- Please ensure the manuscript includes page numbers for easier referencing and review.
- In the "Introduction" section, to enhance the review of "numerical simulations for pervious concrete," I recommend citing recent publications (e.g., [10.1016/j.mechrescom.2021.103791](https://doi.org/10.1016/j.mechrescom.2021.103791); [10.1016/j.cscm.2022.e00946](https://doi.org/10.1016/j.cscm.2022.e00946)) related to FEM or multiscale modeling of pervious concrete strength.
- On page 9, section 2.3, the presentation of the ANN architecture with 3 hidden layers and 128 neurons prior to the data collection section is puzzling, as this architecture seems similar to certain situations of the ANN model presented in this paper.
- Moving to page 10, section 3.1, we have queries concerning the parameter of static pre-compaction ranging from 50 kN to 300 kN. In actuality, the role of compaction energy for pervious concrete and its associated machine learning models are thoroughly discussed in Le et al. 2022 ([10.1007/s12205-022-1918-z](https://doi.org/10.1007/s12205-022-1918-z)), providing a clear review of this topic. Moreover, the compaction levels, as suggested in ASTM C1688/C1688M, are derived from Sumanasooriya et al. 2011 ([10.1016/j.cemconcomp.2011.06.002](https://doi.org/10.1016/j.cemconcomp.2011.06.002)) and several subsequent sources. It remains unclear why the authors chose these values corresponding to a steel wheel roller instead of the typically used average compaction energy. It would be beneficial to cite arguments supporting this choice, for instance, by referencing relevant literature.
- The two most influential parameters characterizing pervious concrete are typically the water/binder ratio and aggregate size. However, in this paper, these parameters are held constant. The authors might consider addressing this in their analysis.
- The experimental data for pervious concrete is known to be highly diverse due to the influence of pore structure. Notably, this paper's results are based on only 9 samples with two independent inputs. It would be beneficial for the authors to comment on this limitation. They could consider utilizing larger data collections available in existing literature, such as those from Zhang et al. 2020 - [10.1016/j.conbuildmat.2020.118803](https://doi.org/10.1016/j.conbuildmat.2020.118803) or Le et al. 2023 - [10.1007/s13369-023-08396-2](https://doi.org/10.1007/s13369-023-08396-2), to enhance the robustness of their findings.
- In Figures 8 and 9, it would be valuable for the author to specify the type of dataset used, whether it's the training set, testing set, or the entire dataset. Furthermore, clarification on the physical indicators represented in Figure 8 would enhance the figure's comprehensibility.
- Figure 10 stands as a pivotal result in this paper. From our interpretation, the traditional ANN is applied to the experimental dataset consisting of 9 samples as input, while the KS-ANN is utilized for data enrichment. Additionally, the representation of the three outputs in one figure, accompanied by a single R2 value, implies that the ANN model is the same for all three outputs (strength, permeability, and porosity). The authors might consider addressing this in their analysis.
- Figures 12 and 14 could benefit from an elaboration on the significance of "different random

numbers" controlling the model's output predictions. A more detailed explanation in this regard would enhance the reader's understanding of the model's behavior and results. The authors might consider addressing this in their analysis.

Authors' Responses to Ms. Ref. No. NCOMMS-23-36290A-Z Reviewer Comments

Reviewers' comments:

Reviewer #1:

This paper presents a combined method of real laboratory tests, mathematical modeling, and machine learning using a kriging-based surrogate model-aided artificial neural network (KS-ANN) to reduce laboratory testing costs and improve predictive accuracy. The proposed method was verified by limited test data of permeable cement-stabilized base materials (PCBMs). In general, this research gives a significant reference for rapid prediction of some required test parameters when the related sampling size is small due to limited test cost and time. Meanwhile, the manuscript is organized well with acceptable readability as well as originality and novelty. However, some following concerns should be discussed if considered for publication,

[Authors' response]: First of all, thank you for reviewing this manuscript and sharing your valuable comments, which have significantly improved the quality of our manuscript. We sincerely appreciate your recognition of our research work presented. As you may find, all your review comments have been addressed one by one with itemized responses. It is our hope that the revised manuscript would be up to your satisfaction. Our aim was to ensure that the revised manuscript meets the rigorous requirements of Nature Communications journal. If you have any further questions for us to clarify, please do not hesitate to let us know. We are definitely committed to promptly addressing any concerns you may have.

(1) The proposed KS-ANN framework is a general method or algorithm to do the accurate extending prediction for the small size of test sampling data. The prediction of PCBMs' behavior is just a case study for verification of this method. Therefore, the question is how to understand the focus of this research. For the current title, it is recommended to be revised to emphasize the general method while not the PCBMs.

[Authors' response]: Thank you very much for the careful review and valuable comments. We acknowledge that the KS-ANN framework was not highlighted prominently or sufficiently in the original manuscript due to length limit. As per your constructive comments, we have thoroughly discussed and carefully re-considered the title and have revised it to "A Novel Kriging-based Surrogate Model Aided Artificial Neural Network (KS-ANN) — with Application to predicting Mechanical Strength and Permeability of Permeable Cement-Stabilized Base Materials". In this revised title, we emphasize the KS-ANN framework and present the prediction of the mechanical strength and permeability of PCBMs as a case study or application example. We hope that the revised title has addressed your concerns adequately.

(2) The mechanical strength and permeability of PCBMs are easy to test in practice, so another question is the necessity for the application of this proposed method. Please enhance the background to explain the reason for verifying the KS-ANN method using PCBMs or consider more appropriate applications like some complicated laboratory tests as the case recommendation.

[Authors' response]: Thank you very much for the careful review and valuable comments. In fact, PCBMs are similar to pervious concrete and present challenges in balancing mechanical strength and permeability properties during material design phase. Extensive laboratory studies have been conducted by previous scholars, such as Xie et al. (2020), Sandoval et al. (2017, 2019, 2020a, 2020b), Zhong & Wille (2016), Deo & Neithalath (2010), and Wang et al. (2020), to investigate the relationship between compressive strength and permeable pore structure in pervious concrete. They also sought to establish predictive models for such relationship between strength and pervious pore. These studies yielded significant advancements and provided valuable guidance for optimizing the design of pervious concrete. However, laboratory test methods are time-consuming, labor-intensive, and costly in nature. When changes occur in actual engineering conditions, conducting laboratory performance evaluation tests of materials requires considerable economic costs and labor. In light of this deficiency, several researchers have explored alternative methods, such as random lattice discrete particle methods (Fascetti et al., 2022), discrete element methods (Cavalaro et al., 2019; Pieralisi et al., 2021; Wang et al., 2022), finite element methods (Rodrigues et al., 2018; Huang et al., 2022), CFD methods (Pieralisi et al., 2017), and self-developed numerical simulation methods (Vu et al., 2021; Nguyen et al., 2022). These methods have been utilized to evaluate the mechanical strength and infiltration properties of pervious concrete, aiming to reduce testing costs and

expedite the design process. The balance between the mechanical strength and permeability properties of permeable concrete is a significant concern. To address this, researchers are exploring the use of data-driven machine learning-based methods to assess these properties instantly, thus reducing testing costs and improving design efficiency. Currently, machine learning models are primarily trained by extensive laboratory experimental data to optimize the design of high-performance permeable concrete (Zhang et al. 2020; Le et al. 2023). However, building these models requires a substantial amount of experimental data, leading to dramatic time and economic costs.

In this paper, we proposed the KS-ANN model to address the aforementioned issues. Our model aims to reduce the cost of laboratory testing and expedite the optimal design of materials. Additionally, it is important to note that PCBM has a more intricate preparation process and internal structure than pervious concrete. Pervious concrete typically contains a cement content ranging from 20% to 40% and is self-compacted or compacted using lower forces. Due to the use of relatively higher cement content, pervious concrete generally exhibits greater mechanical strength. In contrast, PCBM usually contains a cement content of 3% to 15% and is compacted using forces ranging from 300 kN to 400 kN. The use of lower cement content in PCBM results in reduced mechanical strength, while the use of higher compaction force may lead to inadequate permeability. Therefore, as compared to pervious concrete, PCBM exhibits a more intricate balancing relationship between mechanical strength and permeability. Moreover, the optimal design of PCBM should be determined by predicting their hydro-mechanical properties, such as mechanical strength, porosity, and coefficient of permeability, based on laboratory design parameters (e.g., cement content and compaction force). However, there currently lacks related research studies in this area. In our research study, we aim to design a high-standard PCBM with maintainable permeability and adjustable mechanical properties. Therefore, the utilization of the PCBM as a case study or real-world application example to validate the proposed KS-ANN model is deemed highly representative and suitable. As you noted, as part of our research work, we are investigating the optimal design of cement-stabilized recycled aggregate permeable base materials using the KS-ANN framework. Our future objective is to maximize the usage (or relative content) of recycled aggregates from construction and demolition wastes in PCBM, while ensuring that such designed PCBM materials satisfy the required mechanical strength and permeability standards. Currently, this research work has confronted several challenges, while the KS-ANN framework has played a significant role in addressing and tackling these challenges. Additionally, we plan to supplement our findings with additional data from laboratory tests, numerical simulations, and related studies in the literature, which will further demonstrate the effectiveness and efficiency of the proposed KS-ANN framework in practical engineering applications. We hope that your concerns have been addressed properly.

To avoid any further confusions, we have added the aforementioned explanations in the introduction section of the revised version, as quoted below for your review.

Pages 4 to 5, Lines 90 to 127:

“PCBM is similar to pervious concrete in terms of the fact that it also confronts challenges in balancing mechanical strength and permeability properties during the design phase. Existing studies mainly focused on disclosing the correlation mechanisms between compressive strength of pervious concrete and its pervious pore structures, as well as on developing predictive models that relate strength properties with pervious pore characteristics. Such work was proven valuable in guiding the optimal design of pervious concrete¹⁻⁸. However, laboratory testing methods are known to be time-consuming, labor-intensive, and economically costly; further, when changes occur in actual engineering conditions, extensive laboratory tests are required to re-evaluate hydro-mechanical material properties, causing significant financial and labor investments. In light of such deficiency, a variety of alternative methods were employed to evaluate mechanical strength and infiltration properties of pervious concrete, including the random lattice discrete particle method⁹, discrete element method (DEM)¹⁰⁻¹², finite element method (FEM)^{13,14}, computational fluid dynamics (CFD) method¹⁵, and other self-developed numerical simulation methods¹⁶⁻¹⁸. The aim of those methods is to reduce the expenses associated with laboratory tests and expedite the material design process. The balance between mechanical strength and permeability properties of pervious concrete is a crucial area of study. By utilizing data-driven machine learning-based methods, it becomes possible to efficiently evaluate these properties and minimize testing costs while enhancing design efficiency. Currently, machine learning models are primarily trained by extensive laboratory test data to expedite the development of high-performance permeable concrete designs^{19,20}. However, the development of machine learning models requires substantial datasets, leading to much greater data demands and economic costs. To overcome these challenges, there is a need to create a cost-effective machine learning model that can streamline experimentation and expedite the design of materials.

As compared to pervious concrete, PCBM involves more intricate preparation processes and internal structures. Pervious concrete generally contains 20% to 40% cement^{21,22} and is self-compacted or compacted by lower compaction forces^{18,19,23}. Due to relatively higher cement content, pervious concrete usually exhibits greater mechanical strength; conversely, PCBM typically contains 3% to 15% cement²⁴ and is compacted by forces up to 300 kN. The reduced cement content leads to decreased mechanical strength, while increased compaction force may result in inadequate permeability of the PCBM. Xiao et al. (2021) established a preliminary linkage between field and laboratory material compaction by directly applying field compaction force levels in laboratory specimen compaction²⁵. However, there still lacks in-depth research in this area specifically for permeable cement-stabilized base materials. As compared to

pervious concrete, PCBM exhibit much more intricate balance between mechanical strength and permeability properties. The prediction of PCBM properties, such as mechanical strength, porosity, and coefficient of permeability, from laboratory design parameters (e.g., cement content and compaction force) are highly indispensable for optimizing PCBM design. However, few studies have focused on this specific aspect.”

References

1. Xie, X.G., Zhang, T.S., Wang, C., Yang Y.M., Bogush, A., Khayrulina, E., Huang, Z.M., Wei, J.X., Yu, Q.J. Mixture proportion design of pervious concrete based on the relationships between fundamental properties and skeleton structures. *Cement and Concrete Composites* **113**, 103693 (2020).
2. Sandoval, G. F., Galobardes, I., Teixeira, R. & Toralles, B. M. Comparison between the falling head and the constant head permeability tests to assess the permeability coefficient of sustainable Pervious Concretes. (2017) doi:10.1016/J.CSCM.2017.09.001.
3. sandoval, G. F. B., reyes, I. G., Schwantes-Cezario, N., Moura, A. C. & Toralles, B. M. Correlation between Permeability and Porosity for Pervious Concrete (PC). *DYNA* (2019) doi:10.15446/DYNA.V86N209.77613.
4. Sandoval, G. F., Moura, A. C. D., Jussiani, E., Andrello, A. & Toralles, B. M. Proposal of maintenance methodology for pervious concrete (PC) after the phenomenon of clogging. (2020) doi:10.1016/j.conbuildmat.2020.118672.
5. Sandoval, G. F., Galobardes, I., Campos, A. & Toralles, B. M. Assessing the phenomenon of clogging of pervious concrete (Pc): Experimental test and model proposition. (2020) doi:10.1016/j.job.2020.101203.
6. Zhong, R. & Wille, K. Linking pore system characteristics to the compressive behavior of pervious concrete. (2016) doi:10.1016/J.CEMCONCOMP.2016.03.016.
7. Deo, O. & Neithalath, N. Compressive behavior of pervious concretes and a quantification of the influence of random pore structure features. *Materials Science and Engineering: A* **528**, 402–412 (2010).
8. Wang, Z., Zou, D., Liu, T., Zhou, A. & Shen, M. A novel method to predict the mesostructure and performance of pervious concrete. *Construction and Building Materials* **263**, 120117 (2020).
9. Fascetti, A., Ichimaru, S. & Bolander, J. E. Stochastic lattice discrete particle modeling of fracture in pervious concrete. *Computer-Aided Civil and Infrastructure Engineering* **n/a**, (2022).
10. Cavalaro, S. H. P., Blanco, A. & Perialisi, R. Holistic modelling approach for special concrete: from fresh- to hardened-state. *RILEM Tech Lett* **3**, 84–90 (2019).
11. Perialisi, R., Cavalaro, S. H. P. & Aguado, A. Discrete element modelling of mechanical behaviour of pervious concrete. *Cement and Concrete Composites* **119**, 104005 (2021).
12. Wang, X.M., Xiao, Y.J., Shi, W.B., Ren, J.J., Liang, F., Lu J.Q., Li, H., Yu, X.X. Forensic analysis and

numerical simulation of a catastrophic landslide of dissolved and fractured rock slope subject to underground mining. *Landslides* **19**, 1045–1067 (2022).

13. Rodrigues, E. A., Manzoli, O. L., Bitencourt, L. A. G., Bittencourt, T. N. & Sánchez, M. An adaptive concurrent multiscale model for concrete based on coupling finite elements. *Computer Methods in Applied Mechanics and Engineering* **328**, 26–46 (2018).
14. Huang, Y., Yang, Z., Zhang, H. & Natarajan, S. A phase-field cohesive zone model integrated with cell-based smoothed finite element method for quasi-brittle fracture simulations of concrete at mesoscale. *Computer Methods in Applied Mechanics and Engineering* **396**, 115074 (2022).
15. Pieralisi, R., Cavalaro, S. H. P. & Aguado, A. Advanced numerical assessment of the permeability of pervious concrete. *Cement and Concrete Research* **102**, 149–160 (2017).
16. Nguyen, H.-Q., Tran, B.-V. & Vu, T.-S. Numerical approach to predict the flexural damage behavior of pervious concrete. *Case Studies in Construction Materials* **16**, e00946 (2022).
17. Vu, V.-H., Tran, B.-V., Le, B.-A. & Nguyen, H.-Q. Prediction of the relationship between strength and porosity of pervious concrete: A micromechanical investigation. *Mechanics Research Communications* **118**, 103791 (2021).
18. Sumanasooriya, M. S. & Neithalath, N. Pore structure features of pervious concretes proportioned for desired porosities and their performance prediction. *Cement and Concrete Composites* **33**, 778–787 (2011).
19. Le, B.-A., Tran, B.-V., Vu, T.-S., Vu, V.-H. & Nguyen, V.-H. Predicting the Compressive Strength of Pervious Cement Concrete based on Fast Genetic Programming Method. *Arab J Sci Eng* (2023) doi:10.1007/s13369-023-08396-2.
20. Zhang, J., Huang, Y., Ma, G., Sun, J. & Nener, B. A metaheuristic-optimized multi-output model for predicting multiple properties of pervious concrete. *Construction and Building Materials* **249**, 118803 (2020).
21. Pieralisi, R., Cavalaro, S. H. P. & Aguado, A. Discrete element modelling of the fresh state behavior of pervious concrete. *Cement and Concrete Research* **90**, 6–18 (2016).
22. Martins Filho, S. T., Pieralisi, R. & Lofrano, F. C. Framework to characterize nonlinear flow through pervious concrete. *Cement and Concrete Research* **151**, 106633 (2022).
23. Zhao, X., Dong, Q., Chen, X., Han, H. & Zhang, T. Evaluation of fatigue performance of cement-treated composites based on residual strength through discrete element method. *Constr. Build. Mater.* **306**, 124904 (2021).
24. Dong, Q., Zheng, D., Zhao, X., Chen, X. & Chen, Y. Mesoscale numerical simulation of fracture of cement treated base material during semi circular bending test with discrete element model. *Construction and Building Materials* **261**, 119981 (2020).
25. Xiao, Y.J., Wang, M., Wang, X.M., Ren, J.J., Wang, W.D., Chen, X.B. Evaluating Gyrotory Compaction Characteristics of Unbound Permeable Aggregate Base Materials from Meso-Scale Particle Movement Measured by Smart Sensing Technology. *Materials* **14**, 4287 (2021).

(3) The basic idea of this research could be summarized in two steps. The first is that a limited number of lab test results were used to obtain the statistical distribution parameters of the input and output variables. Then, Monte Carlo sampling methods were applied to generate more sample data to train the ANN model. Here, the question is that the ANN model is trained by virtual data generated by the Monte Carlo method even though the limited laboratory data, as we know, any model is better trained by real data, so how to explain the reliability of the prediction by the trained ANN model in theoretical view?

[Authors' response]: Thank you very much for the careful review and valuable comments. As you pointed out, it is widely acknowledged that training ANN models with real data can yield better results. However, it is important to realize the limitations associated with this approach. Obtaining a machine learning model with high accuracy requires a large number of real data samples, which can be both costly and time-consuming. This poses great challenges for the rapid evaluation and design of our research targets. In particular, studies related to large-scale projects (e.g., bridge failure and landslide hazard assessments) often involve the scarcity of real field data. Therefore, to assess bridge failures (Ma et al., 2022; García-Macías & Ubertini, 2022) and landslide hazards (Zeng et al., 2021; Sun et al., 2021), both Monte Carlo and surrogate modeling-based approaches were employed. Such approaches involve the use of a fuzzy input method and ensure the stability of the mean and variance when limited real data are used. Additionally, extensive laboratory test data on mechanical strength and permeability of pervious concrete (Pieralisi et al., 2017, 2021; Vu et al., 2021; Nguyen et al., 2022) demonstrated that even under the same material preparation conditions, there exists a certain level of dispersion in the obtained test data. Therefore, we abandoned the conventional approach of increasing sample size and instead employed the Markov Chain Monte Carlo (MCMC) method and the kriging-based surrogate (KS) model to augment our actual data. Such enriched data, in theory, maintained the same mean and variance as the original data, thus preserving the distributional characteristics of the real data. Consequently, by inputting these expanded datasets into the ANN model, the true distribution characteristics of the real data can be effectively determined, resulting in accurate prediction outcomes. The accuracy of this approach is demonstrated comparable to that of training the ANN model with a substantial amount of genuine test data. Hence, the proposed KS-ANN model holds theoretical reliability; besides, the KS-ANN model reduces the economic and time costs that are typically associated with conducting a large number of laboratory tests. Additionally, this model has great potential in real-world engineering applications. We hope that your concerns have been addressed adequately.

References

1. Ma, Y.-Z. et al. Adaptive Kriging-based failure probability estimation for multiple responses. *Reliability Engineering & System Safety* 228, 108771 (2022).
2. García-Macías, E. & Ubertini, F. Real-time Bayesian damage identification enabled by sparse PCE-Kriging meta-modelling for continuous SHM of large-scale civil engineering structures. *Journal of Building Engineering* 59, 105004 (2022).
3. Zeng, P., Sun, X., Xu, Q., Li, T. & Zhang, T. 3D probabilistic landslide run-out hazard evaluation for quantitative risk assessment purposes. *Engineering Geology* 293, 106303 (2021).
4. Sun, X. et al. From probabilistic back analyses to probabilistic run-out predictions of landslides: A case study of Heifangtai terrace, Gansu Province, China. *Engineering Geology* 280, 105950 (2021).
5. Pieralisi, R., Cavalaro, S. H. P. & Aguado, A. Discrete element modelling of mechanical behaviour of pervious concrete. *Cement and Concrete Composites* 119, 104005 (2021).
6. Pieralisi, R., Cavalaro, S. H. P. & Aguado, A. Advanced numerical assessment of the permeability of pervious concrete. *Cement and Concrete Research* 102, 149–160 (2017).
7. Vu, V.-H., Tran, B.-V., Le, B.-A. & Nguyen, H.-Q. Prediction of the relationship between strength and porosity of pervious concrete: A micromechanical investigation. *Mechanics Research Communications* 118, 103791 (2021).
8. Nguyen, B. T., Ishikawa, T., Zhu, Y., Siva Subramanian, S. & Nguyen, T.-T. New simplified transient method for determining the coefficient of permeability of unsaturated soil. *Engineering Geology* 300, 106564 (2022).

(4) Some more background on the KS model could be provided to explain the conflict between highly accurate predictions and strong divergence indicated in Line 105 to Line 106. In addition, the Monte Carlo can generate samples based on the distribution function of data in general. What is the purpose of using Markov chain? Please clarify.

[Authors' response]: Thank you very much for the careful review and valuable comments. We sincerely apologize for the confusion caused by the lack of detailed explanation of the term 'strong divergence'. Zeng et al. (2021) and Sun et al. (2021) demonstrated that the prediction results of the KS model do not converge once they exceed the domain of the real data. Consequently, to ensure convergence, they established a domain for the monitoring data used in the KS model for the purpose of landslide hazard assessment. Subsequently, they employed the KS model to generate virtual data within this domain. We also

incorporated the KS model into our research, following the aforementioned guidelines. As noted by Zeng et al. (2021), pure Monte Carlo methods may experience convergence issues. On the other hand, the Markov chain Monte Carlo (MCMC) method overcomes this problem by generating samples through random wandering on a Markov chain. This introduces correlation between the samples, leading to improved accuracy of the estimates and faster convergence. Therefore, the purpose of using Markov chains in the original manuscript is to maximize the convergence of the estimated data.

We acknowledge that the description of this process in our original manuscript lacked sufficient details, and we deeply apologize for this careless mistake. In the revised manuscript, we have added necessary descriptions of the background knowledge, as quoted below for your review. We hope that your concerns have been addressed adequately, and we are committed to addressing any further concerns you may have.

Page 7, Lines 182 to 185:

“However, according to the previous studies by Zeng et al. (2021)¹ and Sun et al. (2021)², the prediction results of the KS model may not converge outside the domain of the real data.”

References

1. Zeng, P., Sun, X., Xu, Q., Li, T. & Zhang, T. 3D probabilistic landslide run-out hazard evaluation for quantitative risk assessment purposes. *Engineering Geology* 293, 106303 (2021).
2. Sun, X. Zeng, P., Li, T., Wang, S., et al. From probabilistic back analyses to probabilistic run-out predictions of landslides: A case study of Heifangtai terrace, Gansu Province, China. *Engineering Geology* 280, 105950 (2021).

(5) For the case study of PCBMs, the real test data from three different static compaction forces, 100, 150, and 200 kN, and the compaction force of 300 kN is used for prediction. The case is okay, but some other cases in the output are nonlinear or complicated concerning the input in different regions may not be suitable. Please explain.

[Authors' response]: Thank you very much for the careful review and valuable comments. The proposed KS-ANN model in the original manuscript considered different combinations of cement content and compaction force as input parameters, while the unconfined compressive strength, porosity, and coefficient of permeability are regarded as output parameters. The main objective of KS-ANN was to address the issue of limited real data available from laboratory tests, which can significantly reduce testing costs and thus

enhance design efficiency. Hence, the KS-ANN model is applicable wherever the inputs and outputs are suitable for traditional machine learning models. For example, in our recent study, we investigated the particle crushing value (CV) of a mixture formed using natural gravel (NG), recycled mortar (RM), and recycled brick (RB) particles, and recommended the maximum addition proportions of recycled aggregates (RM and RB) for different base types. To achieve this, we utilized the KS-ANN model with relative proportions of NG, RM, and RB particles as inputs and CV as the output. The results obtained from the model provided recommendations for the maximum proportion of recycled aggregates for different highway grades and base types. These findings are visually summarized in Figures 27 and 28 for illustration purpose. We hope that your concerns have been addressed adequately, and we are committed to addressing any further concerns you may have.

Fig. 27. Design paradigm for the maximum proportions of different types of aggregates studied under compaction forces of (a) 400 kN and (b) 300 kN

Fig. 28. Design ternary diagram for recycled aggregate mixtures studied under compaction forces of (a) 400 kN and (b) 300 kN

(6) It is better to compare the difference in accuracy between a model trained on the expanded dataset and a model trained on the experimental dataset. Moreover, validation should be performed on real data to verify the applicability of the model in the real world.

[Authors' response]: Thank you very much for the careful review and valuable comments. Indeed, we fully agree with what you emphasized. As you noted, to validate the effectiveness of the proposed KS-ANN model, we enumerated the validation results on both extended dataset and real dataset. The results on the extended dataset are shown in Fig. 8b, indicating the coefficient of determination (R^2) value of 0.94. Similarly, the validation results of the trained KS-ANN model on the realistic dataset are presented in Fig. 10b, revealing the R^2 value of 0.92. This improvement in prediction accuracy by 21% over that of the traditional ANN model (shown in Fig. 10a) demonstrates the efficacy and promising potential of the proposed KS-ANN model. These results further confirm the applicability and accuracy of the model in real-world scenarios. We hope that your concerns have been addressed adequately, and we are committed to addressing any further concerns you may have.

b. Actual prediction results for all physical indicator

Fig. 8 Prediction results of the KS-ANN model on the real laboratory test dataset

Fig. 10 Prediction results obtained by both the traditional ANN model and the proposed KS-ANN model

(7) Are variables related to working conditions also random variables with specific distributions? How do these conditions correspond to the physical parameters sampled from the distribution? How can it be verified that such a correspondence is consistent with the actual conditions?

[Authors' response]: Thank you very much for the careful review and valuable comments. In this paper, the design parameters do not follow a specific mathematical/statistical distribution. When investigating the impact of design parameters (i.e., cement content and compaction force) on material properties such as mechanical strength, porosity, and coefficient of permeability, it is important to tailor the design parameters based on the intended use of the material and the corresponding specifications. In this study, the compaction force was set at three different levels: 100 kN, 150 kN, and 200 kN. This approach allows for the examination of material performance under various design parameters. However, laboratory test results often exhibit dispersity or randomness under defined working conditions. In cases where there is an adequate number of replicate specimens, a single set of design parameters may correspond to multiple test results, which can be attributed to the variations in the internal structures of individual specimens. Consequently, a set of design parameters correspond to a statistical distribution of a specific test result rather than a single value. Therefore, we indeed considered in our study this correspondence from the laboratory test setup, which is consistent with the actual situation. We hope that your concerns have been addressed adequately, and we are committed to addressing any further concerns you may have.

(8) From this research, "a statistical distribution model is established using actual laboratory test data. It is used to generate input parameters for Markov Chain Monte Carlo (MCMC) simulations and a kriging-based

surrogate model" If the lab test results can be used to obtain the statistical distribution, what is the purpose of using the KS model? Please clarify. It seems the ANN can be built without any input from the KS. The key is the MC sampling.

[Authors' response]: Thank you very much for the careful review and valuable comments. As you noted, laboratory test results were used to construct and fit statistical distribution models for the unconfined compressive strength (σ), porosity (P), and coefficient of permeability (K), whereas MCMC simulations were employed to generate a large number of extended datasets based on the developed statistical distribution models. However, Fig. 1d shows that the MCMC simulation increased the amount of data for a test results (e.g., unconfined compressive strength (σ), porosity (P) and permeability coefficient (K)). On the other hand, Fig. 1e reveals that the purpose of the KS model is to increase the number of design parameter combinations (e.g., cement content (C) and compaction force (F)). Kriging-based surrogate models can predict physical indexes (σ , P, K) for different design parameter combinations (C, F). Therefore, the complete dataset is considered enriched only after performing surrogate modeling based on kriging. In essence, MCMC increases the dataset of test results for a single design parameter combination, while KS increases the number of design parameter combinations across the entire space of interest. The KS model relies on the generation of the MCMC dataset as its inputs. Consequently, the output of the KS model comprises a vast array of working conditions and the corresponding statistical distribution of physical indicators for each condition. By utilizing this output as the input for the ANN model, a robust ANN model can be trained. It is evident that relying solely on the MCMC results as input for the ANN would lead to a limited amount of working conditions and hinder the accurate prediction of physical indicator for various combinations of cement content and compaction force. We hope that the above responses have addressed your concerns adequately. If you have any further questions or require further clarifications, we are committed to getting them addressed properly once again.

Fig. 1 Proposed framework for using a kriging-based surrogate (KS) model to aid ANNs (KS-ANN) for improving the prediction accuracy of the PCMB's strength and coefficient of permeability: (a) laboratory testing procedure; (b) statistical distribution models constructed using laboratory test results; (c) MCMC simulations; (d) the KS model built; (e) KS model simulated results; and (f) the network structure of the ANN model

(9) Are the MCMC simulations of the three physical parameters performed separately? If so, how was the correspondence between the simulation results determined, i.e., how were the 3D coordinates of each point in Figure 6 determined? The markers of the points in Figure 6 are difficult to distinguish and need to be modified.

[Authors' response]: Thank you very much for the careful review and valuable comments. As you noted, in the original manuscript, we conducted separate MCMC simulations for the three physical parameters to establish direct linkages between material design parameters (i.e., cement content and compaction force) and physical performance indicators (unconfined compressive strength, porosity, and coefficient of permeability), respectively. Previous studies by Vu et al. (2021) and Nguyen et al. (2022) confirmed the

relationship between the unconfined compressive strength and porosity of pervious concrete, while Perialisi et al. (2017) established the linkage between the porosity and coefficient of permeability. Therefore, once the design parameters are determined, the mechanical strength, porosity, and coefficient of permeability exhibit a certain corresponding relationship, and MCMC simulations do not alter this relationship. Furthermore, the simulation results of the MCMC method in our manuscript provide the design parameters corresponding to the physical performance indicators. The aim was to predict three important physical performance indicators using two key design parameters. In Figure 6, the xyz coordinates represent the porosity (P), coefficient of permeability (K), and unconfined compressive strength (σ). The graph shows the representations of the porosity (P), coefficient of permeability (K), and unconfined compressive strength (σ) under different combinations of design parameters. The purpose was to observe the spatial distributions of the physical performance indicators corresponding to different combinations of design parameters. For example, as for the specimen denoted as “5-100” (i.e., with a cement content of 5% and a compaction force of 100 kN), the porosity (P) and coefficient of permeability (K) exhibit wider distributions than does the unconfined compressive strength. However, for the specimen denoted as “20-200”, the opposite pattern is observed. It has been noted that the data points in Figure 6 are labeled too small, making them difficult to distinguish. To address this, a random selection of some MCMC result data was used to redraw Figure 6, ensuring that the selected data adequately represent the spatial distribution characteristics. The modified results of Figure 6 are shown below. We hope that the above responses have addressed your concerns adequately. If you have any further questions or require further clarifications, we are committed to getting them addressed properly once again.

Fig. 6 MCMC simulation results

“Figure 6 presents the MCMC simulation results. To make the labels in the figure clearly visible, we randomly selected a small number of MCMC result data points for plotting. It is worth noting that the randomly selected data points represent the spatial distribution characteristics of all the data.”

References

1. Vu, V.-H., Tran, B.-V., Le, B.-A. & Nguyen, H.-Q. Prediction of the relationship between strength and porosity of pervious concrete: A micromechanical investigation. *Mechanics Research Communications* 118, 103791 (2021).
2. Nguyen, B. T., Ishikawa, T., Zhu, Y., Siva Subramanian, S. & Nguyen, T.-T. New simplified transient method for determining the coefficient of permeability of unsaturated soil. *Engineering Geology* 300, 106564 (2022).
3. Perialisi, R., Cavalaro, S. H. P. & Aguado, A. Advanced numerical assessment of the permeability of pervious concrete. *Cement and Concrete Research* 102, 149–160 (2017).

(10) Compared to transfer learning, what are the significant advantages of KS-ANN? In addition, with the current rise of big model approaches to learning by migration, does virtual data based on agent models have the potential to serve as the underlying data for migration learning? Please supplement some more information into the literature review.

[Authors' response]: Thank you very much for the careful review and valuable comments. As you noted, the weight files of well-trained large models become increasingly important with the increasing use of migration learning in recent years. However, obtaining satisfactory weight files for large models is possible only when large datasets are available. In the field of civil engineering, which relies on expensive laboratory tests, acquiring such datasets is challenging. The concept of the KS-ANN offers a promising solution to this problem, because it requires only a small amount of data to construct statistical distributions, resulting in the creation of large extended datasets. Based on this, excellent weight files of large models can be trained and used as a foundation for transfer learning. We appreciate your valuable comments, as they have sparked our interest in expanding our work on KS-ANN. In the future, we aim to utilize the KS-ANN to address various prediction problems in the field and develop customized large models whenever possible. Additionally, we will provide base weight files for other researchers in the field to utilize for transfer learning. Furthermore, as per your suggestion, we have included necessary additional descriptions in the literature review section, as quoted below for your review. We hope that the above responses have addressed

your concerns adequately. If you have any further questions or require further clarifications, we are committed to getting them addressed properly once again.

Page 3, Lines 61 to 69:

“Further, the popularity of transfer learning has led to the need for a large amount of raw datasets to train valuable weight files of transfer learning¹⁻⁴. However, obtaining such datasets is often challenging in the civil engineering field due to the time-consuming and labor-intensive nature of extensive complex laboratory tests involved. In contrast, a machine learning paradigm based on kriging-based surrogate modeling aided artificial neural networks can be used to enrich limited datasets. This approach provides valuable references for addressing this challenge and effectively reduces the cost of constructing weighting models of migration learning.”

References

1. Su, J., Yu, X., Wang, X., Wang, Z. & Chao, G. Enhanced transfer learning with data augmentation. *Engineering Applications of Artificial Intelligence* **129**, 107602 (2024).
2. Cole, J. M. A Design-to-Device Pipeline for Data-Driven Materials Discovery. *Acc. Chem. Res.* **53**, 599–610 (2020).
3. Qu, T., Zhao, J., Guan, S. & Feng, Y. T. Data-driven multiscale modelling of granular materials via knowledge transfer and sharing. *International Journal of Plasticity* **171**, 103786 (2023).
4. Valikhani, Jaber, Pouyanfar, Mantawy, & Azizinamini. Machine learning and image processing approaches for estimating concrete surface roughness using basic cameras. *Computer-Aided Civil and Infrastructure Engineering* **36**, 213–226 (2021).

(11) Please explain the difference between Fig 8 (a) and (b). Meanwhile, some of the data labels in Figures were very hard to read. Please check and improve the quality of the charts.

[Authors' response]: Thank you very much for the careful review and valuable comments. In contrast to conventional data processing, the test data under specific working conditions in this study exhibit statistical distributions. It is customary to calculate the average of these test results to obtain a representative value. However, in our research work presented, each specific operating condition yielded multiple discrete test results following statistical distributions. Consequently, we utilized the mean values of various physical performance indicators from the KS model outputs as the test dataset in Fig. 8a, and incorporated discrete data from different physical performance indicators as the test dataset in Fig. 8b. The results demonstrate that the KS-ANN model accurately predicts the mean values of the data output from the KS model. These

findings confirm the stability of the unbiased estimation of the KS model in surrogate modeling and validate the robustness of the KS-ANN model. As detailed previously, we have already addressed the lack of clarity in the description of Figure 8 in the revised manuscript. Furthermore, to enhance the visibility and distinguishability of the labels in the figure, we have reduced the amount of data used for plotting. The improved figure with visible labels is quoted below for your review. We hope that the above responses have addressed your concerns adequately. If you have any further questions or require further clarifications, we are committed to getting them addressed properly once again.

a. Prediction of the mean values of each physical performance indicator

b. Actual prediction results for all physical performance indicator

Fig. 8 Prediction results of the KS-ANN model on the test dataset

Page 23, Lines 520 to 529:

“To evaluate the robustness of the KS-ANN model, a test dataset was created using the mean values of various physical performance indicators output by the KS model (see Fig. 8a). Additionally, another test dataset was generated using discrete data of different physical performance indicators output by the KS model (see Fig. 8b). These test datasets were used to verify the performance of the KS-ANN model. The results in Figure 8a demonstrate that the KS-ANN model accurately predicts the mean of the data output based on the KS model, with the coefficient of determination (R^2) value of 0.99. These findings confirm the reliability of the unbiased estimation of the KS model in surrogate modeling. Moreover, the results in Figure 8b indicate that the KS-ANN model results in the R^2 value of 0.94 when predicting the discrete distribution of data, further supporting the robustness of the KS-ANN model.”

Reviewer #2:

This paper aims to construct a Kriging-Based Surrogate Model combined with ANNs using experimental data to predict key properties of porous cement-based concrete materials. We strongly believe that this is the first instance of a data enrichment technique empowering a machine learning model applied to conventional pervious concrete. While it presents interesting aspects, there are certain unclear points that require revision, as outlined below:

[Authors' response]: Thank you very much for the careful review and valuable comments, as your positive compliment and constructive comments not only encouraged us but also significantly contributed to improving the quality of our manuscript. We are grateful to your pointing out that there were still some unclear points in our original manuscript that required further revisions. As per your suggestions, we have thoroughly revised and improved the manuscript, as you may find from the revised version. We hope that the revised version would not only address your concerns and clear your doubts, but also would satisfy the rigorous requirements and high standards of this prestigious journal. Once again, we sincerely thank you for your review comments. If you have any further questions or require any further clarifications, we are committed to doing so wherever necessary.

- Please ensure the manuscript includes page numbers for easier referencing and review.

[Authors' response]: Thank you very much for the careful review and valuable comments. We sincerely apologize for this careless mistake. In the revised version, we have ensured that both page and line numbers are included properly.

- In the "Introduction" section, to enhance the review of "numerical simulations for pervious concrete," I recommend citing recent publications (e.g., [10.1016/j.mechrescom.2021.103791](https://doi.org/10.1016/j.mechrescom.2021.103791); [10.1016/j.cscm.2022.e00946](https://doi.org/10.1016/j.cscm.2022.e00946)) related to FEM or multiscale modeling of pervious concrete strength.

[Authors' response]: Thank you very much for the careful review and valuable comments. We sincerely apologize for missing some of the latest publications on finite element or multiscale modeling of pervious concrete. As per your recommendation, we have thoroughly reviewed the pertinent studies in the literature. For instance, Vu et al. (2021) focused on the homogenization technique and its application in constructing a prediction model for the strength and porosity of pervious concrete. Their research work provides a reliable and accurate method for assessing the strength and permeability of pervious concrete in engineering practice. Nguyen et al. (2022) further developed a numerical model that incorporates the statistical

distribution of a concrete structure. Their model specifically aims to analyze the correlation between the flexural strength and porosity of pervious concrete, allowing for a better understanding of its flexural damage behavior. The study also highlighted the importance of pore structure in determining the mechanical strength of pervious concrete. This finding emphasizes the need to consider the pore morphology during the design stage to meet the desired mechanical strength requirements. In addition, the above references are relevant to the literature review section of our manuscript in terms of numerical simulation methods for pervious concrete, providing valuable insights into the nonlinear damage process and void blocking mechanism of permeable cement-stabilized base materials for our future studies. Therefore, we have cited those relevant studies in the introduction section of the revised manuscript, as quoted below for your review. Once again, we sincerely thank you for your review comments. If you have any further questions or require any further clarifications, we are committed to doing so wherever necessary.

Pages 4, Lines 98 to 104:

“In light of such deficiency, a variety of alternative methods were employed to evaluate mechanical strength and infiltration properties of pervious concrete, including the random lattice discrete particle method¹, discrete element method (DEM)²⁻⁴, finite element method (FEM)^{5,6}, computational fluid dynamics (CFD) method⁷ and other self-developed numerical simulation methods⁸⁻¹⁰. The aim of those methods is to reduce the expenses associated with laboratory tests and expedite the material design process.”

References

1. Fascetti, A., Ichimaru, S. & Bolander, J. E. Stochastic lattice discrete particle modeling of fracture in pervious concrete. *Computer-Aided Civil and Infrastructure Engineering* n/a, (2022).
2. Cavalaro, S. H. P., Blanco, A. & Pieralisi, R. Holistic modelling approach for special concrete: from fresh- to hardened-state. *RILEM Tech Lett* 3, 84–90 (2019).
3. Pieralisi, R., Cavalaro, S. H. P. & Aguado, A. Discrete element modelling of mechanical behaviour of pervious concrete. *Cement and Concrete Composites* 119, 104005 (2021).
4. Wang Xiaoming et al. Forensic analysis and numerical simulation of a catastrophic landslide of dissolved and fractured rock slope subject to underground mining. *Landslides* 19, 1045–1067 (2022).

5. Rodrigues, E. A., Manzoli, O. L., Bitencourt, L. A. G., Bittencourt, T. N. & Sánchez, M. An adaptive concurrent multiscale model for concrete based on coupling finite elements. *Computer Methods in Applied Mechanics and Engineering* 328, 26–46 (2018).
6. Huang, Y., Yang, Z., Zhang, H. & Natarajan, S. A phase-field cohesive zone model integrated with cell-based smoothed finite element method for quasi-brittle fracture simulations of concrete at mesoscale. *Computer Methods in Applied Mechanics and Engineering* 396, 115074 (2022).
7. Peralisi, R., Cavalaro, S. H. P. & Aguado, A. Advanced numerical assessment of the permeability of pervious concrete. *Cement and Concrete Research* 102, 149–160 (2017).
8. Nguyen, H.-Q., Tran, B.-V. & Vu, T.-S. Numerical approach to predict the flexural damage behavior of pervious concrete. *Case Studies in Construction Materials* 16, e00946 (2022).
9. Vu, V.-H., Tran, B.-V., Le, B.-A. & Nguyen, H.-Q. Prediction of the relationship between strength and porosity of pervious concrete: A micromechanical investigation. *Mechanics Research Communications* 118, 103791 (2021).
10. Sumanasooriya, M. S. & Neithalath, N. Pore structure features of pervious concretes proportioned for desired porosities and their performance prediction. *Cement and Concrete Composites* 33, 778–787 (2011).

- On page 9, section 2.3, the presentation of the ANN architecture with 3 hidden layers and 128 neurons prior to the data collection section is puzzling, as this architecture seems similar to certain situations of the ANN model presented in this paper.

[Authors' response]: Thank you very much for the careful review and valuable comments. We sincerely apologize for the ambiguity and confusion caused by improper organization of certain contexts. As per your constructive comment, we have reconsidered our previous approach of presenting the already tuned ANN model directly in the section prior to data collection. We acknowledge that this may have caused confusion, and we appreciate your kind reminder. After a thorough deliberation, we have decided to relocate the corresponding description part to the beginning of section 4.3, as quoted below for your review. This adjustment is necessary because the data collection procedure was discussed in the preceding section. We hope that this revision addresses your concern and we sincerely thank you for your contribution. If you have any further questions or require any further clarifications, we are committed to doing so wherever necessary.

Page 23, Lines 509 to 514:

“4.3 Analysis of the prediction results of the KS-ANN models

For an ANN model, the greatest challenge is to determine the values of the parameters in the hidden layers. Herein, the trial and error strategy were used to continuously adjust the parameters in the hidden layers to minimize the prediction error of the ANN model. After fine tuning those parameters, the number of hidden layers in the ANN model was determined to be 3, with each hidden layer containing 128 neurons.”

• Moving to page 10, section 3.1, we have queries concerning the parameter of static pre-compaction ranging from 50 kN to 300 kN. In actuality, the role of compaction energy for pervious concrete and its associated machine learning models are thoroughly discussed in Le et al. 2022 (10.1007/s12205-022-1918-z), providing a clear review of this topic. Moreover, the compaction levels, as suggested in ASTM C1688/C1688M, are derived from Sumanasooriya et al. 2011 (10.1016/j.cemconcomp.2011.06.002) and several subsequent sources. It remains unclear why the authors chose these values corresponding to a steel wheel roller instead of the typically used average compaction energy. It would be beneficial to cite arguments supporting this choice, for instance, by referencing relevant literature.

[Authors' response]: Thank you very much for the careful review and valuable comments. As you noted, Le et al. (2022) and Sumanasooriya et al. (2011) extensively discussed the compaction performance of pervious concrete and developed various machine learning models to predict its compressive strength and void ratio. Additionally, thorough investigations have been conducted on the impact of compaction on the pore structure characteristics of pervious concrete. These studies provide valuable references for the construction of our original manuscript. However, unlike pervious concrete, permeable cement-stabilized recycled aggregate base materials require greater compaction force to ensure structural stability. Our in-depth field studies in the real-world engineering construction sites have revealed that in order to prevent large-scale aggregate particle migration and ensure stable adherence of cement paste to the aggregate surfaces, field compaction needs to involve turning off the vibratory compaction portion of the roller and relying solely on static compaction pressure. Therefore, the permeable cement-stabilized recycled aggregate base material was compacted using a maximum static compaction force of 300 kN. Furthermore, the previous study on gyratory compaction utilized applied vertical loads to simulate the compaction force of a real roller (Xiao et al., 2021). Based on this inspiration, we decided to utilize the static compaction force associated with the roller as the compaction force for laboratory sample preparation. This approach allowed us to investigate the changes in the mechanical strength and permeability of the specimens under various compaction pressure conditions, aiming to establish a strong correlation with field compaction scenarios. We sincerely apologize for any inconvenience caused by our careless mistake in not thoroughly reviewing this aspect. As per your recommendation, we have added necessary explanations and justifications in the related section of the revised manuscript, as quoted below for your review. We hope that this revision

addresses your concern and we sincerely thank you for your contribution. If you have any further questions or require any further clarifications, we are committed to doing so wherever necessary.

Pages 4 to 5, Lines 113 to 122:

“As compared to pervious concrete, PCBM involves more intricate preparation processes and internal structures. Pervious concrete generally contains 20% to 40% cement^{1,2} and is self-compacted or compacted by lower compaction forces³⁻⁵. Due to relatively higher cement content, pervious concrete usually exhibits greater mechanical strength; conversely, PCBM typically contains 3% to 15% cement⁶ and is compacted by forces up to 300 kN. The reduced cement content leads to decreased mechanical strength, while increased compaction force may result in inadequate permeability of the PCBM. Xiao et al. (2021) established a preliminary linkage between field and laboratory material compaction by directly applying field compaction force levels in laboratory specimen compaction⁷. However, there still lacks in-depth research in this area specifically for permeable cement-stabilized base materials.”

References

1. Perialisi, R., Cavalaro, S. H. P. & Aguado, A. Discrete element modelling of the fresh state behavior of pervious concrete. *Cement and Concrete Research* 90, 6–18 (2016).
2. Martins Filho, S. T., Perialisi, R. & Lofrano, F. C. Framework to characterize nonlinear flow through pervious concrete. *Cement and Concrete Research* 151, 106633 (2022).
3. Sumanasooriya, M. S. & Neithalath, N. Pore structure features of pervious concretes proportioned for desired porosities and their performance prediction. *Cement and Concrete Composites* 33, 778–787 (2011).
4. Le, B.-A., Tran, B.-V., Vu, T.-S., Vu, V.-H. & Nguyen, V.-H. Predicting the Compressive Strength of Pervious Cement Concrete based on Fast Genetic Programming Method. *Arab J Sci Eng* (2023) doi:10.1007/s13369-023-08396-2.
5. Le, B.-A. et al. Predicting the Compressive Strength and the Effective Porosity of Pervious Concrete Using Machine Learning Methods. *KSCE J Civ Eng* 26, 4664–4679 (2022).
6. Dong, Q., Zheng, D., Zhao, X., Chen, X. & Chen, Y. Mesoscale numerical simulation of fracture of cement treated base material during semi circular bending test with discrete element model. *Construction and Building Materials* 261, 119981 (2020).

7. Yuanjie Xiao et al. Evaluating Gyrotory Compaction Characteristics of Unbound Permeable Aggregate Base Materials from Meso-Scale Particle Movement Measured by Smart Sensing Technology. *Materials* 14, 4287 (2021).

- The two most influential parameters characterizing pervious concrete are typically the water/binder ratio and aggregate size. However, in this paper, these parameters are held constant. The authors might consider addressing this in their analysis.

[Authors' response]: Thank you very much for the careful review and valuable comments. As you noted, both mechanical strength and microscopic and meso-scale structures of pervious concrete are indeed influenced by the water/binder ratio and aggregate size. To understand the impact of these parameters, it is often necessary to conduct laboratory tests with multiple control variables and analyze the underlying mechanisms involved. The focus of this study was to develop a KS-ANN framework that enhances the prediction accuracy for small datasets. As a case study towards potential engineering applications, the prediction of the mechanical strength and permeability of permeable cement-stabilized recycled aggregate base material was selected and examined. The objective of this case study was to establish direct correlations between laboratory design parameters (i.e., only cement content and compaction force at this current stage) and physical performance indicators (i.e., currently unconfined compressive strength, porosity, and coefficient of permeability). The optimal material design parameters were then obtained directly from the prescribed values of the physical performance indicators. In our original manuscript, the parameters of the water/binder ratio and aggregate size were held constant or fixed, of which the explanations and justifications have been added in the revised manuscript, as quoted below for your review.

In fact, our ongoing research work has focused on the complex influencing mechanisms of the water/binder ratio and aggregate size on the mechanical strength and microscopic/meso-scale structures of cement-stabilized recycled aggregate base materials. Subsequent manuscripts underway will present detailed analysis results in such aspects. For instance, our recently completed manuscript emphasizes the quantitative characterization of the microscopic/meso-scale structures of cement bridges and voids within permeable cement-stabilized recycled aggregate base materials, as depicted in the figure below. In essence, we achieved this by combining X-ray CT technique and deep learning algorithms to explore the influences of cement content and compaction force on the spatial distribution of cement bridges, the evolution of void morphology, and the permeability characteristics. We hope that the above responses and the ongoing research work underway would address your concerns and satisfy your requirements. If you have any further questions or require any further clarifications, we are committed to doing so wherever necessary.

Fig. 1 Graphical Abstract of the upcoming article entitled "Quantitative Assessment of Cement Bridges and Voids in Cement-Stabilized Permeable Base Materials Using a Mask R-CNN-Based CT Image Segmentation Strategy"

Page 15, Lines 374 to 375:

“Thus, the range of aggregate sizes used in this study was fixed from 4.75 mm to 26.5 mm, and the water-cement ratio was fixed at 0.4.”

- The experimental data for pervious concrete is known to be highly diverse due to the influence of pore structure. Notably, this paper's results are based on only 9 samples with two independent inputs. It would be beneficial for the authors to comment on this limitation. They could consider utilizing larger data collections available in existing literature, such as those from Zhang et al. 2020 -

10.1016/j.conbuildmat.2020.118803 or Le et al. 2023 - 10.1007/s13369-023-08396-2, to enhance the robustness of their findings.

[Authors' response]: Thank you very much for the careful review and valuable comments. As you pointed out, in the study by Zhang et al. (2020), a meta-heuristic multi-output model was developed to predict the unconfined compressive strength and coefficient of permeability of pervious concrete. Their model utilized the water-cement ratio, aggregate-cement ratio, and aggregate particle size as input parameters. This method is allegedly highly robust and provides valuable guidance for designing high-performance pervious concrete. In another separate study by Le et al. (2023), a superior Fast Genetic Programming method was developed. Their method allegedly allows for the consideration of the target porosity and compressive strength during the design stage of pervious concrete, while maintaining a low computational cost. Such examples of proposed methods in the literature offer theoretical guidance for optimizing the design of pervious concrete and holds significant reference value. Those two references are found quite useful for enhancing the background of our manuscript, and they have been properly cited in the introduction section of our revised manuscript. As per your suggestion, the reliability of our findings can be further explored by utilizing larger datasets from the literature. Nevertheless, the main objective of our manuscript was to address the problem of prediction accuracy for samples with small datasets, and the prediction of the mechanical strength and coefficient of permeability of permeable cement-stabilized recycled aggregate base materials merely serves as a case study to validate the effectiveness of the proposed model in our study. To construct a sample of small datasets, nine working conditions with different combinations of cement content and compaction force were adopted. This is indeed a limitation that should be emphasized in our manuscript, which we have added in the revised version, as quoted below for your review. Additionally, the differences between permeable concrete and permeable cement-stabilized recycled aggregate base materials have led to unclear conversion relationships between the two materials. However, further work is definitely needed to establish these relationships through the construction of the MCMC model and the KS model. Due to space constraints, we plan to apply the KS-ANN model specifically to predict different scenarios in a subsequent study. This upcoming study currently underway will include the datasets from the two references recommended by you and is expected to provide valuable insights for the optimal design of pervious concrete and permeable cement-stabilized recycled aggregate base materials. We hope that the above responses and the ongoing research work underway would address your concerns and satisfy your requirements. If you have any further questions or require any further clarifications, we are committed to doing so wherever necessary.

Pages 13 to 14, Lines 329 to 339:

“To investigate the influences of compaction force and cement content on the strength and coefficient of permeability of PCBMs, this study designed three different levels of static compaction force (i.e., 100 kN, 150 kN, and 200 kN) and three different levels of cement content (i.e., 5%, 10%, and 20%). Therefore, nine different working conditions corresponding to different combinations of cement content and compaction force were specified. The combinations of these parameters yielded the tested unconfined compressive strength (σ), porosity (P), and coefficient of permeability (K), which forming a relatively small dataset. The obtained small datasets were used as inputs for the KS-ANN model, which was trained to predict the strength and coefficient of permeability of PCBMs under the compaction force of 300 kN. Among them, the orthogonal experimental design is shown in Table 1, and 3 replicate specimens were tested for each group of experiments.”

References

1. Zhang, J., Huang, Y., Ma, G., Sun, J. & Nener, B. A metaheuristic-optimized multi-output model for predicting multiple properties of pervious concrete. *Construction and Building Materials* 249, 118803 (2020).
2. Le, B.-A., Tran, B.-V., Vu, T.-S., Vu, V.-H. & Nguyen, V.-H. Predicting the Compressive Strength of Pervious Cement Concrete based on Fast Genetic Programming Method. *Arab J Sci Eng* (2023) doi:10.1007/s13369-023-08396-2.

• In Figures 8 and 9, it would be valuable for the author to specify the type of dataset used, whether it's the training set, testing set, or the entire dataset. Furthermore, clarification on the physical indicators represented in Figure 8 would enhance the figure's comprehensibility.

[Authors' response]: Thank you very much for the careful review and valuable comments. Figures 8 and 9 show the prediction results for the testing dataset. The prediction results for the testing dataset of partial output data's means by the KS model are shown in Fig. 8a, while Fig. 8b displays the prediction results for the testing dataset of discretely distributed data enriched by the KS model. These results provide confirmation of the robustness of the KS-ANN model. For more comprehensive understanding of this section, please refer to the detailed descriptions that we have included in the revised manuscript, as quoted below for your review. The prediction results of each physical performance indicator in Fig. 8b are displayed separately and distinctly in Fig. 9, allowing for a clear observation of each physical performance indicator's prediction results. To enhance the readability of Fig. 8, we have used distinctive symbols for

each physical performance indicator as shown by different legends. Additionally, we have reduced the amount of data in the figure to improve clarity. Furthermore, we have modified Figure 10 in the revised manuscript. The modified results are quoted below for your review. We hope that the above responses have addressed your concerns adequately. If you have any further questions or require further clarifications, we are committed to doing so.

a. Prediction of the mean values of each physical indicator

b. Actual prediction results for all physical indicator

Fig. 8 Prediction results of the KS-ANN model on the test dataset

a. Prediction results of ANN model

b. Prediction results of KS-ANN model

Fig. 10 Prediction results by both the traditional ANN model and the proposed KS-ANN model

Page 23, Lines 520 to 529:

“To evaluate the robustness of the KS-ANN model, a testing dataset was created using the mean values of various physical performance indicators output by the KS model (see Fig. 8a). Additionally, another

testing dataset was generated using discrete data of different physical performance indicators output by the KS model (see Fig. 8b). These testing datasets were used to verify the performance of the KS-ANN model. The results in Figure 8a demonstrate that the KS-ANN model accurately predicts the mean of the data output based on the KS model, with the coefficient of determination (R^2) value of 0.99. These findings confirm the reliability of the unbiased estimation of the KS model in surrogate modeling. Moreover, the results in Figure 8b indicate that the KS-ANN model leads to a the R^2 value of 0.94 when predicting the discrete distribution of data, further supporting the robustness of the KS-ANN model.”

- Figure 10 stands as a pivotal result in this paper. From our interpretation, the traditional ANN is applied to the experimental dataset consisting of 9 samples as input, while the KS-ANN is utilized for data enrichment. Additionally, the representation of the three outputs in one figure, accompanied by a single R^2 value, implies that the ANN model is the same for all three outputs (strength, permeability, and porosity). The authors might consider addressing this in their analysis.

[Authors' response]: Thank you very much for the careful review and valuable comments. In fact, in our manuscript, the prediction results presented in Figure 10 do not utilize the input dataset for any ANN model. Instead, real data from laboratory tests, which were excluded initially, were used. The ANN models for the three outputs (strength, permeability, and porosity) are evidently identical. This similarity is primarily due to the construction of a multi-output ANN model, which is similar to the model proposed by Zhang et al. (2020). We apologize for the lack of details in the description of this part, and we have included more comprehensive explanations in the revised manuscript, as quoted below for your review. We hope that this revision addresses your concern and we sincerely thank you for your contribution. If you have any further questions or require any further clarifications, we are committed to doing so wherever necessary.

Page 8, Lines: 210 to 212:

“Finally, the obtained output data of the KS model were used as a training dataset for the ANN model to train a reliable weight model for predicting the physical performance indicators of the PCBM, including the uniaxial compressive strength (σ), porosity (P), and coefficient of permeability (K). Hence, it is necessary to construct a multi-output artificial neural network (ANN) model to meet the prediction needs of PCBM. Additional details can be found in related work¹.”

References

1. Zhang, J., Huang, Y., Ma, G., Sun, J. & Nener, B. A metaheuristic-optimized multi-output model for predicting multiple properties of pervious concrete. *Construction and Building Materials* 249, 118803 (2020).

- Figures 12 and 14 could benefit from an elaboration on the significance of "different random numbers" controlling the model's output predictions. A more detailed explanation in this regard would enhance the reader's understanding of the model's behavior and results. The authors might consider addressing this in their analysis.

[Authors' response]: Thank you very much for the careful review and valuable comments. As you noted, Figures 12 and 14 benefit from the output of 'different random numbers' generated by the control model. However, we sincerely apologize for not providing further elaboration in this section. In fact, the construction of the KS-ANN model was inspired by the concept of integrated learning (Mienye & Sun, 2022). The MCMC model was constructed using 9 random numbers to generate 9 initial datasets. These datasets had the same mathematical distribution but different data. The 9 initial datasets were then utilized as inputs for the KS model, resulting in 9 different sets of post-enrichment datasets. Finally, the enriched datasets were used as inputs for the 9 ANN models, allowing for the training of 9 distinct weight files. Unlike the true Ensemble Learning approach, the KS-ANN model does not require averaging or voting on different predictions. Instead, it provides a discrete result based on different predictors, aligning with the discrete characteristics of laboratory test results. Moreover, if more discrete data are needed, additional random numbers can be assigned during KS-ANN modeling to achieve this objective. We apologize for not explicitly addressing this in the original manuscript. Thank you for bringing this into our attention. We have included more comprehensive explanations in the revised manuscript. We hope that this revision addresses your concern and we sincerely thank you for your contribution. If you have any further questions or require any further clarifications, we are committed to doing so wherever necessary.

Page: 19, Lines: 450 to 457:

“Furthermore, as suggested by the ensemble learning concept¹, multiple random numbers can be used in MCMC to generate multiple initial datasets with the same mathematical distribution but different data. This allows for training multiple parallel KS-ANN models. The selection of random numbers can be customized based on the specific data requirements. Additionally, the predictions from these models can be averaged or voted to obtain results that meet the desired criteria. Therefore, in this paper, multiple random

numbers were specified in the MCMC simulations to train multiple parallel KS-ANN models for predicting unconfined compressive strength (σ), porosity (P), and coefficient of permeability(K).”

References

1. Mienye, I. D. & Sun, Y. A Survey of Ensemble Learning: Concepts, Algorithms, Applications, and Prospects. *IEEE Access* 10, 99129–99149 (2022).

Editing Comments:

[Authors' response]: Addressed.

We sincerely thank all the reviewers for their insightful and valuable suggestions!

REVIEWERS' COMMENTS

Reviewer #1 (Remarks to the Author):

I have completely checked the revised manuscript, especially the detailed responses. I think the authors did a very good job in careful consideration and revision of what I was concerned about in the original submission of this article. Only one more suggestion is that the limitation of this proposed method in this research could be supplemented in the conclusions or somewhere. In other words, the scope of the application using this method is better clearly stated even if is close to a general method. Except for this, I do not have other suggestions. Congratulations! Hope to see this work published for being a valuable academic contribution to the improvement of the traditional experiment method using advanced algorithms as well as data generating and processing methods.

Reviewer #2 (Remarks to the Author):

Firstly, I am very impressed with the authors' meticulous preparation of the response to reviewers. It seems that the majority of the responses align well with the content of the paper as well as the questions raised by the reviewers.

I only have one query, similar to the first question raised by Reviewer 1. As the authors have emphasized, the main content of this paper revolves around establishing a Novel Kriging-based Surrogate Model Aided Artificial Neural Network (KS-ANN), and the application for Permeable Cement-Stabilized Base Materials is just a case study.

Although the authors have changed the title of the paper to assert their standpoint, the clarification at this point is not entirely clear, and I still lean towards the perspective that the novelty of the paper lies in the application of data enrichment techniques combined with machine learning models for a specific material, rather than purely developing a new technique that can be applied to various types of data problems. Even when the novelty of the paper leans more towards the application of the KS-ANN model for the PCS material, this point still needs to be further clarified.

I agree to accept the paper if the above point is further clarified.

A minor suggestion regarding the compaction force of concrete (line 112): the authors need to adjust the reference to Le, B.-A. (2022) instead of Le, B.-A. (2023), as mentioned in the revised manuscript, to avoid confusion for readers.

Best regards

Authors' Responses to Ms. Ref. No. NCOMMS-23-36290B

Reviewer Comments

Reviewers' comments:

Reviewer #1

I have completely checked the revised manuscript, especially the detailed responses. I think the authors did a very good job in careful consideration and revision of what I was concerned about in the original submission of this article. Only one more suggestion is that the limitation of this proposed method in this research could be supplemented in the conclusions or somewhere. In other words, the scope of the application using this method is better clearly stated even if is close to a general method. Except for this, I do not have other suggestions. Congratulations! Hope to see this work published for being a valuable academic contribution to the improvement of the traditional experiment method using advanced algorithms as well as data generating and processing methods.

[Authors' response]: First of all, we would like to express our sincere gratitude for your careful review of and valuable comments on our revised manuscript. Your recognition of this work has greatly inspired and encouraged us, and we are committed to furthering our efforts, continuing to innovate, and delivering research outcomes of greater value. We fully embrace your suggestions, as it would be objective and beneficial to include the proposed method's applicability in the conclusion. In the latest revised manuscript, we have supplemented this objective applicability scope, excerpted as follows. We hope that the supplementary contents could satisfy your expectation and approval standard.

Page 27, Lines 586 to 590.

(4) The proposed method uses the data-enriching technique assisted by the kriging-based surrogate model, which is highly beneficial for enhancing the predictive accuracy and efficiency of the ANN model. However, to extend the use of this proposed method in other engineering applications, the input data must satisfy the requirements of the kriging-based surrogate models, i.e., they cannot

exceed 4 dimensions. This is the major limitation of this proposed method, which is being tackled in the research efforts currently underway.

Reviewer #2

Dear Editor and Authors,

Firstly, I am very impressed with the authors' meticulous preparation of the response to reviewers. It seems that the majority of the responses align well with the content of the paper as well as the questions raised by the reviewers.

I only have one query, similar to the first question raised by Reviewer 1. As the authors have emphasized, the main content of this paper revolves around establishing a Novel Kriging-based Surrogate Model Aided Artificial Neural Network (KS-ANN), and the application for Permeable Cement-Stabilized Base Materials is just a case study.

Although the authors have changed the title of the paper to assert their standpoint, the clarification at this point is not entirely clear, and I still lean towards the perspective that the novelty of the paper lies in the application of data enrichment techniques combined with machine learning models for a specific material, rather than purely developing a new technique that can be applied to various types of data problems. Even when the novelty of the paper leans more towards the application of the KS-ANN model for the PCS material, this point still needs to be further clarified.

I agree to accept the paper if the above point is further clarified.

A minor suggestion regarding the compaction force of concrete (line 112): the authors need to adjust the reference to Le, B.-A. (2022) instead of Le, B.-A. (2023), as mentioned in the revised manuscript, to avoid confusion for readers.

Best regards

[Authors' response]: First of all, we sincerely appreciate your careful review of our responses to the reviewers' comments and the revised manuscript. Your positive complement is also very encouraging and inspiring, to which we are definitely grateful. As you noted, the novelty of this paper lies in the integration of data-enriching techniques with machine learning models for specific materials rather than merely developing a new technology that can be applied to various types of data or materials. After carefully considering your feedback and incorporating the recommendation

provided by the other reviewer, we have revised the title of our manuscript again to "**Kriging-based Surrogate Data-enriching Artificial Neural Network Prediction of Strength and Permeability of Permeable Cement-stabilized Base**". In the new title, we have effectively showcased the integration of emerging data-enriching techniques with advanced machine learning models, emphasizing the use of the kriging-based surrogate data assisted artificial neural network (KS-ANN) model as an advanced tool for predicting the mechanical strength and permeability of permeable cement-stabilized base materials. According to this new title, we have concurrently updated the abstract of the revised manuscript. Besides, the new title also complies with the requirements of Nature Communications journal for titles not exceeding 15 words. We hope that these modifications could satisfy your expectation and approval standard.

Additionally, we sincerely apologize for the careless citation mistake on line 112. We have now corrected the reference from the erroneous Le, B.-A. (2023) to the correct Le, B.-A. (2022). Thank you once again for your thorough review, and we hope that our revisions are up to your satisfaction.

Pages 1 to 2, Lines 16 to 31.

Limited test data hinder the accurate prediction of mechanical strength and permeability of permeable cement-stabilized base materials (PCBM). In this paper, we developed a kriging-based surrogate model assisted artificial neural network (KS-ANN) framework that integrates laboratory testing, mathematical modeling, and machine learning. A statistical distribution model was established from limited test data to enrich the dataset through the combination of Markov chain Monte Carlo simulation and kriging-based surrogate modeling. Subsequently, an artificial neural network (ANN) model was trained using the enriched dataset. The results demonstrate that the well-trained KS-ANN model effectively captures the actual data distribution characteristics. The accurate prediction of the mechanical strength and permeability of PCBM under the constraint of limited data validates the effectiveness of the proposed framework. As compared to traditional ANN models, the KS-ANN model improves the prediction accuracy of PCBM's mechanical strength by 21%. Based on the accurate prediction of PCBM's mechanical strength and permeability by the KS-ANN model, an optimization function was developed to determine the optimal cement content and compaction force range of PCBM, enabling it to concurrently satisfy the requirements of mechanical strength and permeability. This study provides a reliable, cost-effective, and rapid solution for

evaluating the performance and optimizing the design of PCBM and similar materials.

Page 5, Lines 125 to 128:

“It’s worth noting that the novelty of the study lies in the application of data enrichment techniques combined with machine learning models for the specific PCBMs, rather than purely developing a new technique that can be applied to various types of data problems.”